



# Implementation of mycorrhizal mechanisms into soil carbon model improves the prediction of long-term processes of plant litter decomposition

Weilin Huang[1], Peter M. van Bodegom[1], Toni Viskari[2], Jari Liski[2], and Nadejda A. Soudzilovskaia[1,3]

[1]Environmental Biology, Institute of Environmental Sciences, Leiden University, Einsteinweg 2, 2333CC Leiden, the Netherlands;
[2]Finnish Meteorological Institute, Helsinki, 00101, Finland
[3]Centre for Environmental Sciences, Hasselt University, Martelarenlaan 42, 3500 Hasselt, Belgium

*Correspondence to:* Weilin Huang (w.huang@cml.leidenuniv.nl)

**Abstract**

Ecosystems dominated by plants featuring ectomycorrhizae (EM) and arbuscular mycorrhizae (AM) promote distinct soil carbon dynamics. AM and EM soil environments can thus have different impacts on litter decomposition. However, current soil carbon models treat mycorrhizal impacts on the processes of soil carbon transformation as a black box.

We re-formulated the soil carbon model Yasso15, and incorporated impacts of mycorrhizal vegetation on soil carbon pools of different recalcitrance. We examined alternative conceptualizations of mycorrhizal impacts on transformations of labile and stable carbon, and quantitatively assessed the performance of the selected optimal model in terms of the long-term fate of plant litter.

We found that mycorrhizal impacts on pools of labile carbon in the litter are distinct from that on recalcitrant pools. Plant litter of the same chemical composition decomposes slower when exposed to EM-dominated ecosystems compared to AM-dominated ones, and across time, EM-dominated ecosystems accumulate more recalcitrant residues of non-decomposed litter. Overall, adding our mycorrhizal module into the Yasso model improved the accuracy of the temporal dynamics of carbon sequestration predictions.

Our results suggest that mycorrhizal impacts on litter decomposition are underpinned by distinct decomposition pathways in AM- and EM-dominated ecosystems. Ignoring mycorrhiza-induced mechanisms will thus lead to an overestimation of climate impacts on decomposition dynamics. Our new model provides a benchmark for mechanistic and quantitative modelling of microbial impact on soil carbon. It helps to determine the relative importance of mycorrhizal associations and climate on organic matter decomposition rate and reduces the uncertainties in estimating soil carbon sequestration.

## 1. Introduction

Long-term soil carbon sequestration is to a large extent determined by complex soil-plant rhizosphere and microbial interactions (Dijkstra and Cheng, 2007; Fernandez and Kennedy, 2016; Fontaine et al., 2007; Ostle et al., 2009). These interactions contribute to atmospheric $CO_2$ balance (Ostle et al., 2009; Todd-Brown et al., 2012) and are increasingly recognized as processes that counteract climate change (Terrer et al., 2016). Plant associations with fungi, so-called mycorrhizas, is the most widespread symbiosis on Earth, featured by the majority of vascular plants including trees, shrubs and herbs (Brundrett and Tedersoo, 2018). Mycorrhizae are hypothesized to play especially important roles in soil carbon sequestration, yet the actual mechanisms of mycorrhizal impacts on soil carbon dynamics are poorly understood.

Mycorrhizal fungi themselves are not capable of meaningfully obtaining carbon from decomposing plant litter (Bödeker et al., 2016; Lindahl and Tunlid, 2015). Instead, they receive carbon from their symbiotic host plants. However, the relation between





mycorrhizal fungi and soil C dynamics is enabled through three potential pathways that likely complement each other (Frey, 2019): (i) provisioning of substrate for decomposition (Leake et al., 2004; Soudzilovskaia et al., 2015), (ii) mediating plant litter quality and amounts (Averill et al., 2019; Cornelissen et al., 2001; Phillips et al., 2013), and (iii) controlling the environment of plant litter decomposition, including mediation of the microbial community (Fernandez and Kennedy, 2016; Frey, 2019). The temporal dynamics of plant litter decomposition is underpinned by the dynamics of decomposition of distinct types of carbon-containing molecules (Berg and McClaugherty, 2008; Cornelissen et al., 2007), which could be generally grouped as labile and recalcitrant SOC components. The fate of carbon originating from components of various chemical recalcitrance will ultimately determine the decomposition dynamics (Aponte et al., 2012; Cusack et al., 2009; Kalbitz et al., 2003; McClaugherty et al., 1985). Among the three major pathways of mycorrhizal impacts on soil C dynamic, the pathway of mycorrhizal fungal control on decomposition environment is arguably understood the least.

To understand mycorrhizal fungal impacts on soil carbon dynamics, we need to distinguish between arbuscular mycorrhiza (AM) and ectomycorrhiza (EM) types of symbiosis. Together, these types are possessed by over 80% of plant species compromising the majority of terrestrial plant biomass (Brundrett and Tedersoo, 2018; Soudzilovskaia et al., 2019). While they are present in almost all Earth ecosystems, it has been proposed that distinct mycorrhizal types are associated with specific ecosystems and soil attributes (Craig et al., 2018; Read and Perez-Moreno, 2003; Steidinger et al., 2019). Moreover, distinct mycorrhizal guilds differ in the pathways of nutrient acquisition from decomposing plant litter. AM fungi (AMF) have limited or no ability to depolymerize organic macromolecules. They do not possess enzymes enabling nitrogen extraction and uptake from soil organic matter (Orwin et al., 2011; Treseder et al., 2016; Treseder and Allen, 2002), but primarily acquire inorganic nutrients mobilized by saprotrophic fungi and bacteria. Accordingly, plant litter subjected to AM fungi-dominated decomposition environment is likely to undergo a more balanced decomposition process with both labile and recalcitrant components being degraded by saprotrophic decomposers. On the other hand, compared to AM fungi, most EM fungi (EMF) can produce enzymes involved in decomposing organic compounds of plant litter (Fernandez and Kennedy, 2015; Lindahl and Tunlid, 2015; Zak et al., 2019), and therefore have easier access to organic nutrients, especially so to nitrogen. It has been proposed that EMF increase recalcitrance of decomposing litter, as their ability of nitrogen uptake while withholding carbon compounds from breaking down increases carbon-to-nitrogen ratios in the decomposing plant litter (Nicolás et al., 2019; Read and Perez-Moreno, 2003). This process of gradually increasing recalcitrance of plant litter subjected to EM-dominated decomposition environment is further magnified by the suppression of saprotrophic decomposer activities, the effect known as the Gadgil effect (Fernandez and Kennedy, 2015; Gadgil and Gadgil, 1971; Smith and Wan, 2019). Yet the magnitude of the impacts induced by the differential roles of mycorrhizal types on the recalcitrance and dynamics of decomposing plant litter is understood very poorly, especially so in quantitative terms.

It is crucial to improve the current understanding of the role of mycorrhizas in SOM dynamics (Brzostek et al., 2014; Liang et al., 2017), especially since it is expected to affect modelling of vegetation change impacts on soil-atmospheric carbon exchange (Soudzilovskaia et al., 2019; Steidinger et al., 2019; Terrer et al., 2019). However, traditional field experiments are typically too short to assess the full complexity of the mechanisms underpinning the potential difference of AM and EM impacts on the plant litter decomposition processes over time. Besides, traditional field experiments have limitations in explicitly distinguishing the individual mechanisms of the mycorrhizal impacts of the decomposition process, and are hardly able to assess the impacts of mycorrhizas on the decomposition of distinct chemical fractions of litter. An alternative tool to progress in our understanding of mycorrhizal impacts on plant litter decomposition, is testing different formulations of mycorrhizal impacts in process-based models of litter decomposition, and examining how well the models fit the observations.

Current deterministic models of soil C decomposition (e.g. CENTURY, DAYCENT, DAISY, DNDC, NCSOIL, RothC and Struc-C etc.) do not explicitly account for mycorrhizas as a driver of plant litter decomposition processes. Instead, climate and litter quality, the well-acknowledged regulators of SOC and litter decomposition (Cornwell et al., 2008; Coûteaux et al., 1998;



Cusack et al., 2009; Parton et al., 2007; Zhang et al., 2008) are being modelled as primary drivers of all aspects of SOC dynamics. A body of recent studies have questioned the recognition of climate and litter quality as the only dominant regulators in SOC and litter decomposition (Bradford et al., 2016; García-Palacios et al., 2013; Wall et al., 2008), and plead for explicit inclusion of microbial and especially so mycorrhizal impacts (Johnson et al., 2006; Shi et al., 2016) on soil C dynamics into biogeochemical models (Clemmensen et al., 2013; Craig et al., 2018; Todd-Brown et al., 2012; Wieder et al., 2013). However, 85 so far, models assessing the role of mycorrhizas in SOM dynamics (e.g. Liang et al., 2017; Orwin et al., 2011; Shi et al., 2016) do not compare the relative impacts of mycorrhiza vs. climate on SOM decomposition processes.

In this study, we aim to develop a framework allowing incorporation of mycorrhizal impacts on the decomposition of plant litter into a generic soil C model, specifically addressing one of the most poorly understood mechanisms of mycorrhizal impact on plant litter decomposition – the impact through controlling decomposition environment, separately from climate and other 90 factors. Hereto we focus on answering the following four questions:

- What is the best conceptualization, and accordingly the best representation in a soil C dynamics model, of mycorrhizal impacts on decomposition of plant litter labile and recalcitrant carbon compounds?

- To what extent does modelling mycorrhiza-associated mechanisms in mediating litter decomposition process improve model performance, in terms of model errors and temporal dynamics?

- What is the sensitivity of model prediction to the uncertainty of parameters and input describing the mechanics of decomposition as affected by mycorrhiza vs climate and other factors?

- How are plant litter decomposition patterns affected both in terms of total C loss and loss of C from compounds of distinct recalcitrance by AMF- and EMF-dominated decomposition environments?

## 2. Methods

Among available models of plant litter decomposition, the Yasso model (Tuomi et al., 2011a) provides an ideal framework for a mechanistic integration of mycorrhizal impacts into the modelling of plant litter decomposition processes. Yasso is among the models that underpin IPCC predictions of impacts of environmental change scenarios on global C cycles (IPCC, 2006; IPCC, 2019), which has been widely applied in global process estimations (Steidinger et al., 2019; Tuomi et al., 2009). In the Yasso model, the plant litter decomposition process is presented as a classification of the organic matter into five compartments, 105 characterized on the measurable basis of a chemical solubility of organic matter (Liski et al., 2005): compounds soluble in water (W), carbon compounds hydrolysable in acid (denoted with A), components soluble in a non-polar solvent, e.g. ethanol or dichloromethane (E), compounds neither soluble nor hydrolysable (N), and humus (H) (Berg and Agren, 1984; Palosuo et al., 2005). The W, A and E pools together form the group of labile C fractions of soil organic matter, N a recalcitrant but yet not a mineral bound C fraction, and the H pool represents a fraction of very stable soil C that remains in the soil for decades 110 or centuries.

Figure 1 presents the schematic representation of the Yasso model, with carbon flows quantified from results of the original Yasso model formulation (Tuomi et al., 2011a; Viskari et al., 2020). H pool-related flows are not specified in this figure, because humus can only be produced in deeper soil accessible to mineral compounds, thus is not considered in this study of 10-years litter decomposition simulations. This model presents the litter decomposition process as a system of linear 115 differential equations, and the total amount of carbon released from each pool is the result of flux transferred between pools and C released to the atmosphere as carbon dioxide. Detailed descriptions of the original Yasso model and the dataset used for its parametrization are provided in Appendices A and B, respectively.




The conceptualization of litter decomposition, as the process of C conversion into compartments representing measurable C fractions, makes Yasso a particularly suitable model for the mechanistic modelling of plant litter decomposition process,

allowing new (in our case, mycorrhizal) pathways to be incorporated in a truly mechanistic way.

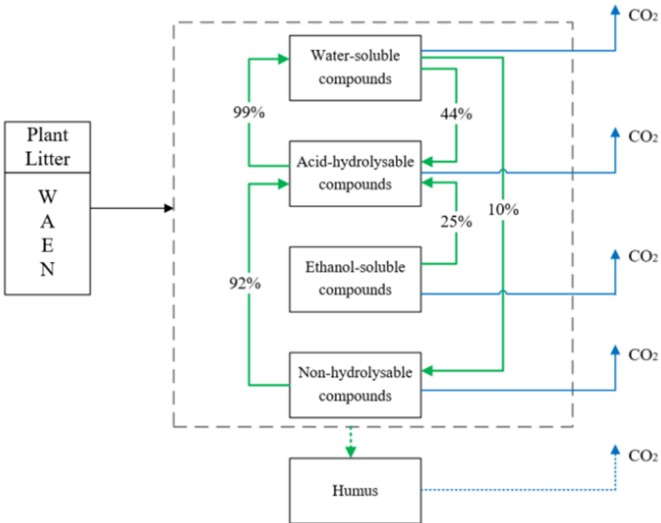

Fig.1 Conceptual diagram of decomposition and mass flows between five carbon pools in Yasso. Conceptual diagram of carbon pools and fluxes in original Yasso model (Tuomi et al., 2011). The fate of organic matter entering soil as plant litter material is represented as a series of carbon fluxes between carbon pools characterized by distinct decomposability (i.e chemical solubility) levels. Values in arrows show the amount of C transformed between pools and leaving the pools according to the original Yasso formulation and parameterizations (Tuomi, et al., 2011; Viskari et al., 2020).


**2.1 Implementation of mycorrhizal impacts on decomposition in Yasso: general principles and data**

We modified the Yasso model by adding mycorrhiza as a factor controlling plant litter decomposition processes. Our model focuses on explaining the fate of aboveground plant litter that is decomposed at the topsoil layer before entering into deeper

mineral soil or subsoil. During this stage, decomposers pre-process plant litter, liberating carbon compounds which – in a later stage- contribute to the accumulation of mineral associated organic matter MAOM and particulate organic matter (POM) through different pathways in deeper soils (Bradford et al., 2016; Cotrufo et al., 2013, 2015, 2019; Sokol et al., 2019).

We conceptualized the process of soil organic matter decomposition as being controlled by the factors currently accounted by Yasso (decomposition environment of temperature and moisture, and litter chemical composition), and additionally as being

dependent on the mycorrhizal environment. We modelled impacts of the mycorrhizal environment on plant litter decomposition, as the sum of impacts caused by the predominance of AM and EM fungal types. As there is no data currently available about the global distribution of mycorrhizal fungal biomass, we approximated the AM and EM fungal biomass to be proportional to the amount of C obtained from each type of mycorrhizal vegetation through photosynthesis. Thus AM and EM fungal biomass were estimated as products of proportions of AM and EM plant biomass in vegetation, and vegetation Gross

Primary Production (GPP, using MODIS product-MOD17 data) (Running et al., 2004; Zhao et al., 2005). Further, each of the AM and EM fungal impacts depends on the fungal-type-specific ability to affect the litter decomposition process.

We parameterized our new model using litter decomposition databases (Appendix B) used in Yasso modelling (Tuomi et al., 2009, 2011b, 2011a): CIDET with the measurements from Canada (Trofymow, 1998), LIDET with data from the USA and Central America (Gholz et al., 2000) and Eurodeco (ED) with data gathered from several European research projects (Berg et

al., 1991). For each site in these datasets, climate and chemical composition data were supplemented with information on the fractions of AM and EM vegetation within total plant biomass, which was extracted from the global mycorrhizal distribution





map of Soudzilovskaia et al. (2019). To avoid potential mismatches between the actual fractions of AM and EM plants within the total plant biomass and the (generalized) data of AM and EM fractions derived from the map of Soudzilovskaia et al. (2019), plant community composition of each site was carefully checked for consistency with the map.

**2.2 Mycorrhizal impact on total decomposition**

We accounted for mycorrhizal impacts in the original Yasso litter decomposition model. Figure 2 shows the general principle of modification of each decomposition pool: the total carbon outflux of each W,A,E,N pool is controlled by two factors: climate (as in the original YASSO model) and mycorrhizal decomposition environment (the new factor added to the model). See Appendix A for details on the decomposition terms used in YASSO.

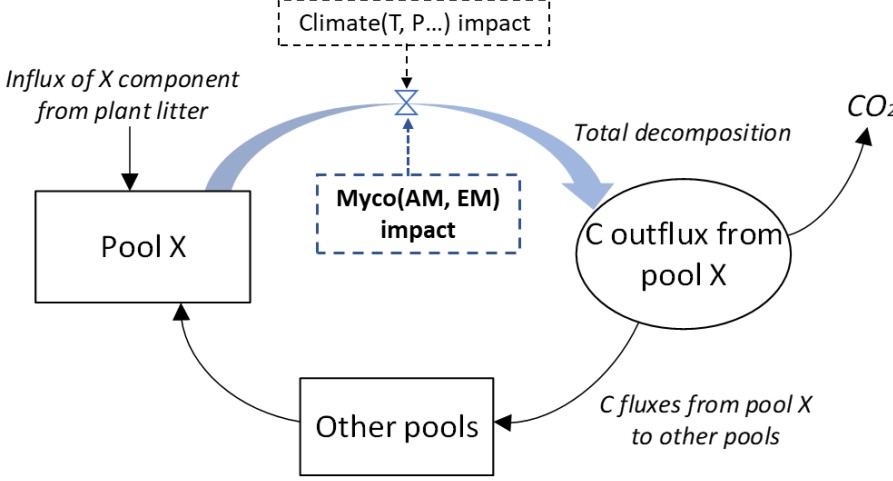


Fig.2 Carbon fluxes from and to each X pool of carbon, with X being W, A, E or N, as represented by the modified Yasso model. Blue arrow and blue box show conceptualization of added impact of mycorrhizal environment on litter decomposition process. While in the original version of the Yasso plant litter decomposition process was represented as a function of climate and litter quality, in our model decomposition is a function of proportions of ectomycorrhizal and arbuscular mycorrhizal plants in vegetation, climate and plant litter quality.

Accordingly, we modified the original form of the equations describing the decomposition rate of each WAEN element in Yasso model. In the original model decomposition matrix (see Appendix A), only climate was considered as a driver of decomposition of AWEN elements in Ki(C) (Ki, where i∈{W,A,E,N}, see A(3) in appendix). In the new model formulation we added a term Mi representing the mycorrhizal impact on the total C outflux of each WAEN pool to Ki(C) Eq. (1):

$$K_i = K_i(\boldsymbol{C})' \cdot (1 + Mi), i \in \{W, A, E, N\} \tag{1}$$

The Mi term is described by Eq. (2):

$$Mi = m_{iAM} \cdot \lambda_{AM} \cdot G_{pp} + m_{iEM} \cdot \lambda_{EM} \cdot G_{pp}, \ i \in \{W, A, E, N\} \tag{2}$$

where $m_{iAM}$ and $m_{iEM}$ are the impacts of AM and EM mycorrhizas on C loss from pool i; $\lambda_{AM}$ and $\lambda_{EM}$ are the fractions of AM and EM vegetation within the total vegetation biomass; $G_{pp}$ is the gross primary production of mycorrhizal vegetation.

We compared four different conceptualizations of AM and EM impacts on distinct WAEN pools of decomposing litter, by

evaluating the performance of four distinct model versions (Fig.3):

Myco-Yasso.v1 – a model the magnitude of mycorrhizal impact on carbon loss from each of the W, A, E, and N pools differs among the pools (Fig.3a);





Myco-Yasso.v2 – a model where mycorrhizal impacts on carbon loss from labile soil C pools (W, E, and A) are equal among the pools, while the mycorrhizal impact on carbon loss from the recalcitrant soil C pool (N) differs from the impact on C losses from labile pools (Fig.3b);

Myco-Yasso.v3 – a model where mycorrhizal impacts on carbon loss are equal for all pools (Fig.3c);

Myco-Yasso.v4 – a model where mycorrhiza affects only carbon loss from the recalcitrant soil C pool (N) (Fig.3d).

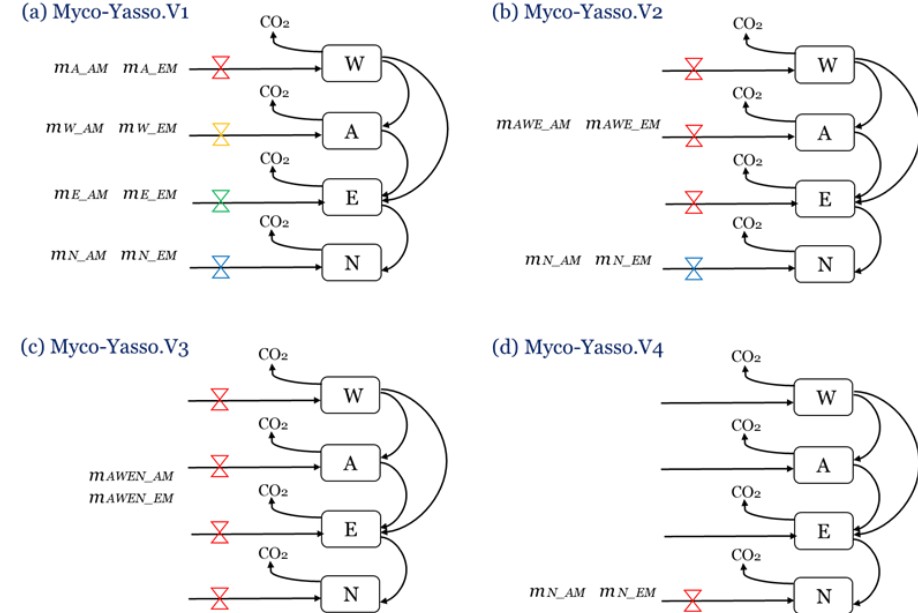

Fig.3 Four conceptualizations of the possible mechanisms of mycorrhizal impacts on litter decomposition, modelled with four versions of the Myco-Yasso model. (a) Myco-Yasso.V1: mycorrhizal impacts differ for each of the W, A, E, and N pools; (b) Myco-Yasso.V2: mycorrhizal impacts on W, A, E pools have the same magnitude, but the mycorrhizal impact on N pool is different; Myco-Yasso.V3: mycorrhizal impacts on W, A, E pools and N pools are equal; Myco-Yasso.V4: mycorrhizae impact only N pool.

We used a Bayesian framework and a Differential Evolution Markov Chain with snooker updater (DEzs, Braak and Vrugt, 2008) algorithm-Markov Chain Monte Carlo (MCMC) (Haario et al., 2001) for calibrating all the relevant parameters following the original Yasso framework (Tuomi et al., 2011a; Viskari et al., 2020). Essential parameters from the original Yasso and newly derived mycorrhizal dependencies with corresponding symbols and units are explained in Table 1. We allowed $m_{iAM}$ and $m_{iEM}$ to vary from negative to positive values. The only control on priors of $m_{iAM}$ and $m_{iEM}$ is limiting $Mi >$ -1 in Eq.(1) to make the algorithm meaningful. The other parameter priors were adopted according to previous Yasso research (Tuomi et al., 2009). We performed cross-validation for each model, using 80% of data randomly drawn from the dataset for calibration and the remaining 20% of the data used for validation. After parameterization, all model versions were examined for *Pearson's r* and RMSE values of the correlation between the predicted and observed data with both the validation dataset and the full dataset. To account for the fact that the data in the different datasets varied in measurement uncertainty and the number of observations, we opted to compare the performance of models separately for each dataset. We use root mean square error (RMSE), Akaike information criterion (AIC) and Bayesian information criterion (BIC) as the criteria for comparing relative quality of models and thereby selecting the optimal model. The conceptualization with the lowest RMSE, AIC and BIC was selected as the optimal model with best performance.





Table 1. Parameters calibrated for each model version

| Parmeter subset | Parameters | Remark | Unit |
|---|---|---|---|
| Decomposition rate parameters | aW | Decomposition rate parameter of W | $\text{yr}^{-1}$ |
| | aA | Decomposition rate parameter of A | $\text{yr}^{-1}$ |
| | aE | Decomposition rate parameter of E | $\text{yr}^{-1}$ |
| | aN | Decomposition rate parameter of N | $\text{yr}^{-1}$ |
| Mass flow parameters | pWA | Relative mass flows from W to A | - |
| | pWN | Relative mass flows from W to N | - |
| | pEW | Relative mass flows from E to W | - |
| Temperature parameters | b1 | Temperature dependence of W,A,E | $°\text{C}^{-1}$ |
| | b2 | Temperature dependence of W,A,E | $°\text{C}^{-2}$ |
| | bN1 | Temperature dependence of N | $°\text{C}^{-1}$ |
| | bN2 | Temperature dependence of N | $°\text{C}^{-2}$ |
| Precipitation parameters | g | Precipitation dependence of W,A,E | $\text{m yr}^{-1}$ |
| | gN | Precipitation dependence of N | $\text{m yr}^{-1}$ |
| Mycorrhiza parameters | miAM | AM mycorrhiza dependence of each pool | $\text{g m}^{-2}\,\text{yr}^{-1}$ |
| | miEM | EM mycorrhiza dependence of each pool | $\text{g m}^{-2}\,\text{yr}^{-1}$ |

### 2.3 Performance of the selected best mycorrhizal model of soil C sequestration

#### 2.3.1 Model residuals and uncertainty analysis

We examined the residuals (differences between measurements and model predicted litter decomposition) as a function of AM
and EM fractions in the biomass of mycorrhizal vegetation. Then, the uncertainty of the selected Myco-Yasso model was
assessed in two aspects:

(a) *Variability in estimating total C mass loss through litter decomposition*. The variability in the percentage of C mass
remaining after 10 years of litter decomposition, as revealed by the original Yasso model and the selected Myco-Yasso model
was examined by conducting Monte Carlo simulations for a hypothetic site. In line with previous sensitivity tests of Yasso
(Liski *et al.*, 2005), we chose the following input data to represent the conditions of decomposition: mean annual temperature
5.2°C, annual precipitation 840mm. For the Myco-Yasso model, the mycorrhizal impact in Eq.(2) was quantified by assuming
an AM mycorrhizal plant biomass proportion of 38%, EM mycorrhizal plant biomass proportion of 36% and a GPP of
1516g·m⁻²·a⁻¹. We used the following values for the chemical composition of the litter: W fraction- 20.6%, A fraction-43.0%,
E fraction- 8.7% and N fraction- 27.7%. We ran 1000 simulations using parameter values randomly selected from even
distributions of the input parameters within their uncertainty ranges.

(b) *Sensitivity to parameters and input*. With environmental conditions and chemical composition of the litter being the same
as used in part (a), we evaluated the sensitivity of litter decomposition by separately increasing model parameters by 1% and
input values by 1% of variations across the dataset in 10-years model runs. This test was conducted for both the original Yasso
model and the selected Myco-Yasso model.

#### 2.3.2 Temporal dynamics of the model

(1) Model performance over time

We examined the ability of the selected model in predicting litter decomposition at different times following litter input, by
comparing model predictions of C mass remaining to real measurements of the remaining C at the same time moment. The



time slots were classified according to the datasets' characteristics, as litter decomposition measurements for different datasets

were taken at different months in a year.

(2) Mycorrhizal impact on labile and recalcitrant litter pools

To analyze total litter decomposition and the different litter pools, simulations of 10-years litter decomposition were conducted in different mycorrhizal environments with varying AM: EM vegetation biomass proportions. Input values in terms of environmental factors and chemical fractions were set consistent with the standard conditions as used in the sensitivity analysis.

Additionally, we conducted the same simulations to test model consistency using different initial inputs with chemical fractions of typical root and leaf litter (Appendix D).

## 3. Results

### 3.1 Model comparison and selection

For all four model versions examined, the calibration based on the full database including all three different datasets showed

a high correlation between measurements and model predictions, with the Pearson's $r$ being 0.84-0.86 for CIDET, 0.67-0.68 for LIDET, and 0.90-0.91 for ED, and small differences between individual versions of Myco-Yasso models. However, the model RMSE comparisons revealed that the Myco-Yasso V2 provided the strongest RMSE decrease among all Myco-Yasso models compared to the original Yasso15 model. This pattern was consistent through all datasets (Table 2). And the AIC and BIC confirm that the Myco-Yasso V2 has the best performance. Based on the RMSE, AIC and BIC, we selected Myco-Yasso

V2 as a model representing the optimal conceptualization of mycorrhizal impact on plant litter decomposition for later analysis. In this model, the mycorrhizal impact is similar among labile C compounds (WAE) but different for the recalcitrant C compound (N). Hereafter this optimal model is referred as Myco-Yasso. Scatterplots showing model improvement in terms of observed vs predicted values for the Myco-Yasso model compared to the original Yasso15 model are provided in Appendix C. Details of parametrization outcomes of the Myco-Yasso model are provided in Table C1.

Table 2. Model performance criteria of RMSE, AIC and BIC. RMSE (model predictions of total mass remaining in plant litter_units %) of different mycorrhizal model versions for different litter decomposition datasets.

| | | Yasso15 | Myco-Yasso.V1 | **Myco-Yasso.V2** | Myco-Yasso.V3 | Myco-Yasso.V4 |
|---|---|---|---|---|---|---|
| Parameter number | | 16 | 24 | **20** | 18 | 18 |
| RMSE each dataset | CIDET | 10.55 | 10.87 | **10.5** | 11.23 | 10.74 |
| | LIDET | 19.94 | 21.09 | **19.32** | 19.87 | 19.83 |
| | ED | 6.85 | 6.96 | **6.57** | 7.09 | 7.01 |
| AIC | | 20639.13 | 20484.89 | 20464.37 | 20630.29 | 20574.72 |
| BIC | | 41338.45 | 41060.07 | 41003.98 | 41328.30 | 41217.16 |

### 3.2 Model performance across the range of mycorrhizal plant biomass fractions in vegetation

The standardized residuals for the litter decomposition measurements (% of C decomposed from initial plant litter) as a

function of AM and EM fractions in the biomass of mycorrhizal vegetation are shown in Fig.4. Within the 95% probability density covered by $2\sigma$ intervals, model predictions agreed well with measurements across the entire range of fractions of biomass AM and EM plants in vegetation. At low values of the AM fractions (AM<10%) and at high values of EM fractions (EM>85%), the model had relatively large negative residuals, suggesting a lower predictive power in these data groups.





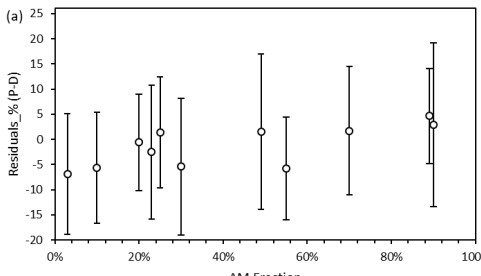 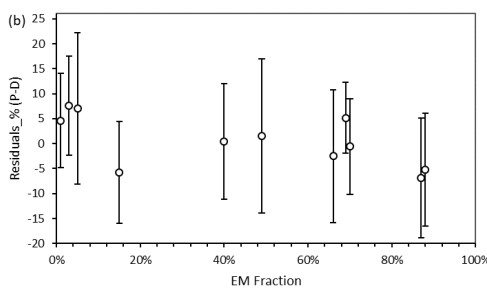

Fig.4 Standardized residuals (Predictions - Measurements, P-M) of C decomposed from initial plant litter (%) modelled by Myco-Yasso as
a function of dominance of AM mycorrhizal plants and EM mycorrhizal plants in vegetation. The dominance level is expressed as: (a)
percentage of AM plant biomass within total plant biomass or (b) percentage of EM plant biomass within total plant biomass. The circle in
the middle of each line is the mean value of the residuals. The intervals contain residuals within 95% probability intervals.

### 3.3 Variability in litter decomposition estimations

The 1000 simulations of the Yasso15 model ran for the conditions of a hypothetical site with prescribed environmental
conditions revealed a normally distributed dataset with $\mu$= 22.56, and $\sigma$= 1.81. The same simulations conducted by the Myco-
Yasso model yielded a dataset with a lower $\mu$ (16.90) and lower $\sigma$ (1.19), indicating a lower total sensitivity of the Myco-Yasso
model to variation in input parameters within the confidence interval. The best-fit normal distributions of these two model
predictions are shown in Fig.5.

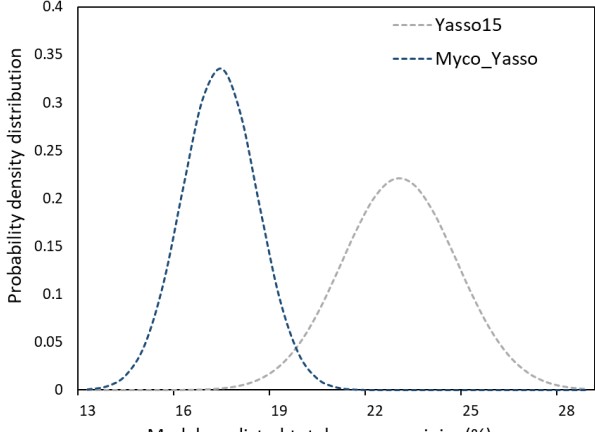

Fig.5 The probability density distribution of litter remaining predicted by Yasso15 compared to Myco_Yasso. Compared to predictions of
Yasso15, Myco-Yasso reduces the variation in the predictions of C mass remaining from decomposing litter after 10-years of decomposition.
The probability density is based on 1000 model runs for conditions of a hypothetical site with the prescribed environmental conditions (see
descriptions in Sect.3.1.1).

**3.4 Sensitivity of litter decomposition to parameters and input values**

The sensitivity of the Myco-Yasso model to the individual litter decomposition parameters is shown in Fig.6. Overall, the
model sensitivity to each subset of parameters decreased (Fig.C3). Model sensitivity to input values is shown in Fig.7. The
magnitude of sensitivity of plant litter decomposition to mycorrhizal impact is comparable to the sensitivity of climate (Fig.6,
Fig.7 and Fig.C3).

The Myco-Yasso model showed the highest sensitivity to the impact of arbuscular mycorrhizal vegetation of the N pool
(mN_AM) out of the four added mycorrhizal impact parameters (Fig.6). This implies that AM environment has a much stronger
stimulating effect on the decomposition of the recalcitrant pool compared to EM environment. In contrast, the decomposition
from the labile pools was only a bit more stimulated by EM environment than by AM environment. Concerning the
decomposition rate parameters, the overall carbon loss in the Myco-Yasso model has a considerably lower sensitivity to the



total decomposition rate of the N pool (αN), and a slightly increased sensitivity to the decomposition rate of the A pool (αA) compared to Yasso15. However, the total impact of all α terms together to the sensitivity of carbon loss prediction is generally similar in Myco-Yasso compared to Yasso15 (Fig.C3).

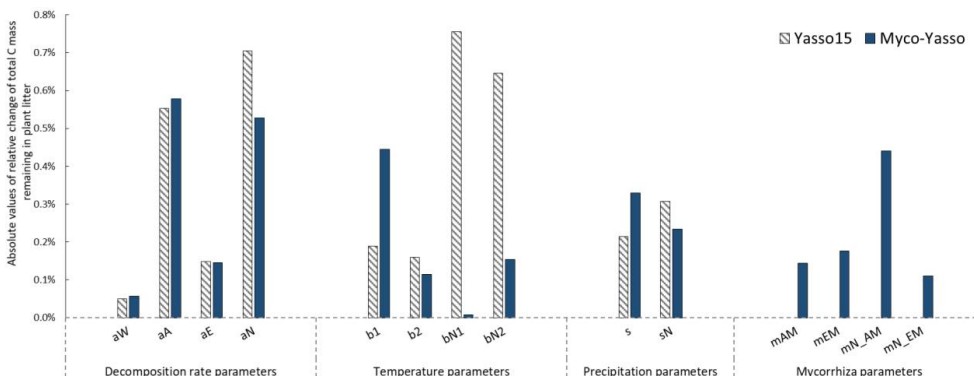

Fig.6 Model sensitivity to 1% increase in individual litter decomposition parameters.

Analysis of the environmental dependencies of the Myco-Yasso (Fig.6) revealed that the new model is less sensitive to the overall variability in temperature parameters ($b1$, $b2$, $bN1$ and $bN2$) than the original Yasso15 model, although the overall effect of temperature sensitivity decrease (Fig.C3) is mostly driven by the decreased sensitivity of N pools to temperature ($bN1$ and $bN2$). The sensitivity to precipitation parameters ($g$ and $gN$) of Myco-Yasso is generally similar to Yasso15, with a slight increase in sensitivity of the WAE pools to precipitation ($g$ parameter) and a slight decrease in sensitivity of N pool to

precipitation ($gN$ parameter). The resulting impacts on carbon transformation in the respective pools are provided in Fig.C4.

Figure 7 shows the model sensitivity to the increase of each input value by 1% of variations across the dataset including initial plant litter quality, climate parameters, and the mycorrhizal environment of decomposition. With an increase in biomass of mycorrhizal plants, less carbon will remain of the decomposing plant litter after 10 years of decomposition. This impact is similar in magnitude to the impact of temperature increase. An increase in EM dominance leads to a slight increase in carbon

accumulation, while AM dominance speeds up decomposition to similar extents as temperature increases. The Myco-Yasso shows a slight decrease in sensitivity to climate variables compared to Yasso15, confirming our supposition that potential mycorrhizal impacts were partly accounted for by climate variables in the original Yasso15.

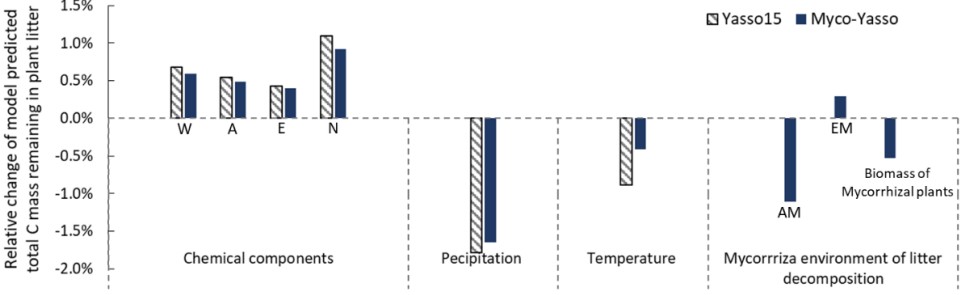

Fig.7 Model sensitivity to 1% (variations) increase in model input values. Impacts of input parameters are shown in terms of the relative change in total C remaining after 10 years of decomposition. The 'AM'-bar shows the impact of an increase of AM plant biomass by 1%, while EM plant biomass remains unchanged; 'EM'-bar shows the impact of an increase of EM plant biomass by 1%, while AM plant biomass remains unchanged; 'Biomass of mycorrhizal plants'-bar shows the impact of an increase of biomass of AM and EM plants together by 1% while the AM and EM distribution within the vegetation remains unchanged.




**3.5 Model predictions of temporal dynamics of plant litter decomposition**

**3.5.1 Model performance over time**

Figure 8 illustrates how the model predictions of the Myco-Yasso improve the modelled decomposition over time compared to the original Yasso15 model using the full dataset. The differences in models' prediction accuracy (RMSE of the Yasso15 predictions minus RMSE of Myco-Yasso predictions) has a trend of increment over time, indicating an increasing impact of mycorrhizas on litter decomposition dynamics across 10 years. The examination with only validation dataset comparing

original model Yasso15 and Yasso-Myco is provided in supplementary material, Fig.C5.

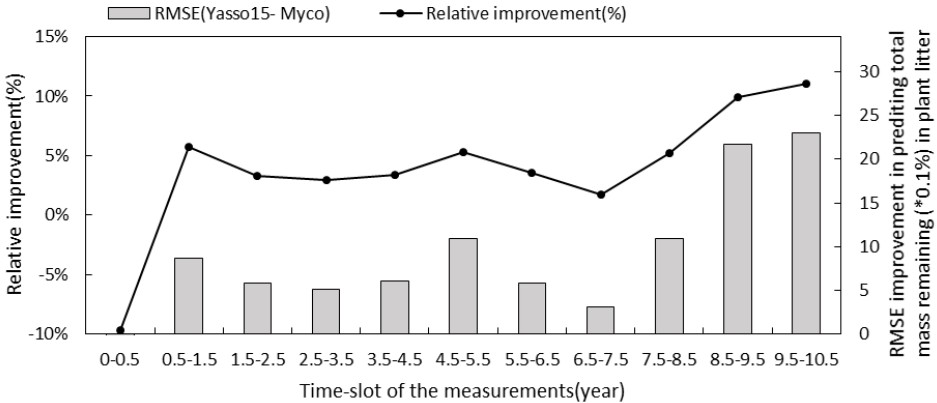

Fig.8 Improvement in Myco-Yasso model accuracy compared to the original Yasso over decomposition period. Bars represent the relative RMSE differences between Yasso15 and Myco-Yasso per period. The line with dots shows the absolute value of the RMSE differences (Yasso15- Myco). Predictions examed from the full dataset simulation.

**3.5.2 Mycorrhizal impact on labile and recalcitrant pools**

Assessments of the dynamics of total litter mass decomposition under the dominance of AM and EM vegetation with Myco-Yasso (Fig.9a) revealed that, at the 10$^{th}$ year of decomposition, plant litter (with equal initial chemical composition) will have ca.15% less carbon remaining if decomposed in an AM-dominated environment compared to an EM-dominated environment. During the 1$^{st}$ decomposition year, litter subjected to AM or EM decomposition environments decomposes with a similar rate,

while at the later stages (after 1 year), litter subjected to an AM environment decomposes faster. The difference in the total mass remaining in an AM vs EM dominant environment increases during the decomposition period from 2-10 years.

Examining the dynamics of carbon loss from distinct individual decomposition compartments (Fig.9b-e) shows that labile carbon components of plant litter (WAE) decompose with a similar rate in AM and EM environments. Recalcitrant carbon compounds of litter (N compounds) tend to accumulate during the first two years. After that, C loss starts to take place in an

EM-dominant environment promoting the accumulation of N components compare to an AM-dominant environment. Comparison among distinct litter types reveals that this pattern is not affected by initial litter quality (Fig.D1 and Fig.D2).

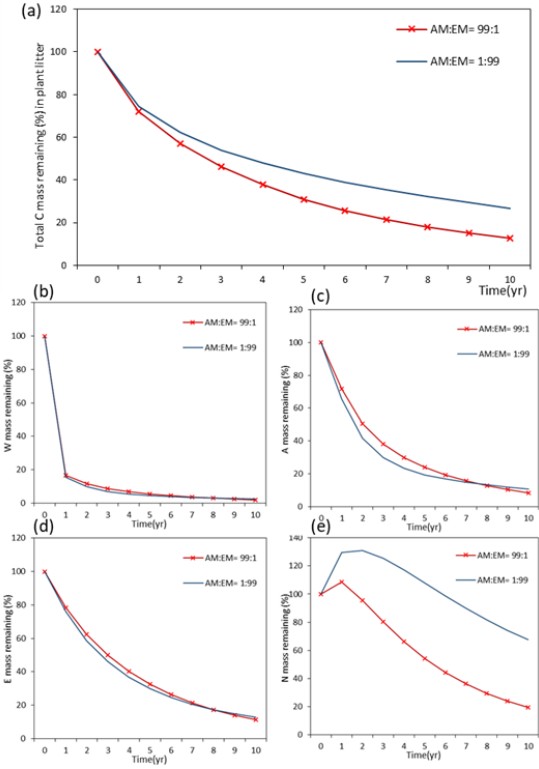

Fig.9 Dynamics of plant litter decomposition in AM dominant vs EM dominant environments. (a) decomposition of total carbon mass from plant litter; (b), (c) and (d) show the dynamics of C remaining of labile carbon components (W – water-soluble C fraction, E – ethanol-soluble C fraction, A – acid hydrolysable C fraction); (e) dynamics of carbon remaining of recalcitrant C component (N- non-hydrolysable fraction).

## 4. Discussion

Mycorrhizal vegetation types are widely recognized to have a strong impact on plant litter decomposition processes and soil carbon pools dynamics. Yet, the mechanisms of mycorrhizal impacts on the soil C cycle are not well-understood, and available data of the relationship between soil C pools and dominance of distinct mycorrhizal types of vegetation are often contrasting each other both at the local (Craig et al., 2018; Phillips et al., 2013) and global scale (Sousssdzilovskaia et al., 2019; Steidinger et al., 2019). The matter is additionally complicated by the fact that mycorrhizas affect C cycles via three mechanistically distinct pathways of (1) provisioning dead mycelium as the substrate for decomposition, (2) mediating plant litter quality and amounts, and (3) controlling the environment of plant litter decomposition. Earlier works did not explicitly differentiate between these pathways (Johnson et al., 2006) or focused mainly on the second pathway (Brzostek et al., 2014). Our study is the first attempt to mechanistically incorporate the impacts of different types of mycorrhizal environment, i.e. the third pathway, into a plant litter decomposition model. Herewith, we explicitly focus on impacts of the mycorrhizal environment on the plant litter decomposition process in topsoil profiles, where plant litter is transformed into soil organic matter and carbon compounds are pre-processed for further potential incorporation into particulate organic matter or minerally-associated (i.e. stable) organic matter. We assessed a full range of concepts representing mycorrhizal impacts on labile and stable components of decomposing litter across a wide range of eco-environmental conditions varying in plant species, litter types and climate variables (Table B1). Overall, the model Myco-Yasso fits the litter decomposition datasets well, especially considering the high level of noise in some of the data and the environmental variation among the sites, in





terms of geology, soil quality and other alike parameters not described by the model. Based on this assessment we provide
insights into the role of distinct mycorrhizal types in long term decomposition processes of labile and recalcitrant
components of plant litter.

### 4.1 Improved representation of temporal dynamics of litter C

The temporal dynamics of organic matter decomposition is among the worst understood aspects of soil C cycling. SOM
decomposition encompasses changes in both the composition of soil C components as well as in their breakdown (García-
Palacios et al., 2016). This duality, in combination with the long term nature of the processes involved, makes experimental
assessments of temporal dynamics of SOM formation to be extremely difficult, and pleas for using modelling approaches.
Incorporation of mycorrhizal impacts into Yasso improved the overall model predictions of soil C in the long-term, indicating
that mycorrhizal impact is a vital factor to be accounted for in analyses of long-term litter decomposition processes, at least in
the topsoil layer. The mycorrhizal impacts are likely less visible in the short-term (< 3 years), and detectable effects of the
mycorrhizal environment of litter should be assessed over a longer period. This is in agreement with earlier studies (e.g.
Paterson *et al.*, 2008) that have shown in short-term [13]C-labelling experiments that labile and recalcitrant plant litter fractions
are utilized by distinct microbial communities, but in the short-term, these communities are not shaped by the presence and
activity of mycorrhizal fungi.

### 4.2 Explicit separation of climate vs mycorrhizal impacts

Our model allows explicit quantification of mycorrhizal impacts on decomposition and separates them from climatic factors.
In the original Yasso model, soil C pools are controlled by litter quality and climate, with the 'climate' factor implicitly
accounting for all global variation of environmental conditions. That original model had high predictive power, especially so
for short-term decomposition processes, to which our re-formulation could provide only an incremental improvement.
However, the oversimplification of the role of climate without considering microbial factors hinders the ability of the models
to examine future impacts of alterations in the climate on soil C dynamics (Pongratz et al., 2018). Such a lack of mechanistic
and quantitative representation of belowground processes is recognized to be a principal source of uncertainty in our
quantifications of global terrestrial biogeochemical cycles (Nyawira et al., 2017; Pongratz et al., 2018; Todd-Brown et al.,
2013; Trumbore, 2006). Hereto our study constitutes a step forward providing a benchmark in the incorporation of microbial
impacts into the modelling of SOC dynamics.

Compared to the original Yasso15 model, the Myco-Yasso model has a lower sensitivity to the total variation in temperature.
The decrease of decomposition sensitivity to temperature suggests that the impact of temperature on decomposition could have
been overestimated in previous global modelling attempts that did not consider mycorrhizae as a driving factor. Undoubtedly,
the temperature regime controls soil and litter respiration (Hobbie, 1996), making the sensitivity to temperature in a soil C
cycle model to be an essential issue for better estimating future soil C stocks change and its feedback to climate. While
modelling approaches allow distinguishing these mechanisms, separation of these two factors from global field observations
is extremely difficult, because of a tight correlation of mycorrhizal distributions to gradients of temperature (Barceló et al.,
2019; Soudzilovskaia et al., 2015).

### 4.3 Mycorrhizal impact on labile litter pools is distinct from that on recalcitrant litter pools

We tested four principally distinct concepts on the impact of the mycorrhizal environment on plant litter decomposition. The
selected model imposes effects of distinct magnitude and direction on both labile and recalcitrant carbon pools. This finding
supports the theory that the turnover of OM depends largely on its composition and recalcitrance of biopolymers (Baldrian,
2017; Berg and McClaugherty, 2008; Cornelissen et al., 2007; Gui et al., 2017), while distinct mycorrhizal types differ in
strategies treating simple organic compounds and recalcitrant compounds (Rajala et al., 2011). This translates into an



accumulation of recalcitrant C components of not-yet decomposed plant litter material in EM-dominated environments, while
AM-dominated environments generally promote decomposition. While the presence of AM does not directly affect
decomposition, the theory that AMF can exert an indirect influence on this process through regulating free-living groups of
decomposers in the soil is well supported. AM fungi alter the physicochemical environment for the microbial community, and
modify the soil bacterial communities (Gui et al., 2017; Nuccio et al., 2013; Offre et al., 2007). AMF stimulate the activity of
particular bacteria (Franco-Correa et al., 2010), which are known to be capable of catalyzing the efficient degradation of labile
and recalcitrant plant litter (Bayer et al., 1998; Kersters et al., 2006). Furthermore, AMF has been shown to prime the
decomposition of organic matter by supplying plant-derived labile C to saprotrophic fungi and bacteria (de Vries & Caruso,
2016), which result in higher microbial turnover and respiration, priming decomposition of SOC, and decreasing the soil C
pool.

In contrast, efficient nutrient uptake by EM fungi promotes immobilization of soil nitrogen in complex organic molecules of
high recalcitrance, and therewith promotes the formation of microbial communities, mostly saprotrophic fungi, able to
decompose such recalcitrant organic substrates (Fernandez and Kennedy, 2016; Langley and Hungate, 2003). While multiple
studies examining the genetic potential of ectomycorrhizal fungi have shown that EM fungi are capable of producing enzymes
degrading complex C and humus (Nicolás et al., 2019), the abundance of such genes is generally low compared to saprotrophic
fungal guilds.

Yet, the question in which direction EM impacts on soil C prevails in the long term has remained unanswered. Similarly, the
long term impacts of AM fungi on saprotrophic communities have to our knowledge been never evaluated quantitatively. Our
modelling exercise provides mechanistic and quantitatively examination on the long-term consequences of the differential
mechanisms of AM and EM impacts on soil C, and suggest that more C is conserved in an EM-dominant environment than an
AM environment particularly due to the accumulation of recalcitrant carbon compounds (independent of the associated litter
quality). More intriguingly, we show that the long-term impacts of both types of mycorrhizas on labile carbon components are
similar.

**4.4 Future improvements of mycorrhizal impacts of SOC modelling**

Our model greatly improves the accuracy of SOM dynamics over time even though we assessed the litter decomposition
process in topsoil profiles across 10-year only. Formation of the most recalcitrant compartment of soil, defined by Yasso model
as "humus" (Tuomi et al., 2011a) is not examined in our study, because we assumed that a 10-year period of litter
decomposition for which we had detailed data for model calibration, was not long enough for forming humus. Future work
should aim at including mycorrhizal impacts on the humus formation process, linking short- and medium-term decomposition
processes to the ultra-long SOM dynamics.

Furthermore, our current work examines the dynamics of SOM in terms of labile and stable compounds, yet not addressing
the fate of stable, minerally-associated soil C, the ultimate pool of soil-sequestered C. During the last decade, the question of
whether the minerally-associated soil C originates from labile C components, possibly undergoing microbial transformation
(Cotrufo et al., 2015, 2019; Mambelli et al., 2011) or occurs through direct sorption of poorly decomposed plant compounds,
was intensively debated (Bradford et al., 2016; Sokol et al., 2019). Recent research (Sokol et al., 2019) has proposed that both
pathways are possible, depending on the capability of the environment to support the release of labile C compounds. While
our work does not address the pathway of formation of minerally stabilized carbon, it provides insights into the important
processes preceding C mineral stabilization, as we examine the long term processes of formation of labile C pools potentially
available for microbial uptake and the development of recalcitrant plant litter pools potentially forming MAOM by binding to
mineral particles. Our study suggests that the EM-dominated decomposition environment tends to promote the accumulation
of poorly decomposed plant compounds supporting the pathway of minerally-associated soil C from undecomposed plant





material, which suggests that EM- and AM-dominated ecosystems differ in POM and MAOM fractions contributed to the process of further SOM decomposition. The question of to what extent this pathway dominates the entire flux of soil C into the pool of minerally-associated C needs to be further evaluated. Such evaluation should additionally consider the processes omitted in this study such as fluxes of labile C from the root and fungal exudates and C fluxes originating from the decomposition of dead mycelium of mycorrhizal fungi (Baskaran et al., 2017; See et al., 2021).

**5. Conclusions**

Our study is the first attempt of modelling the impacts of the mycorrhizal environment on soil carbon decomposition based on the carbon release of specific soil chemical components within pathways of respiration and mass transformation. While mycorrhizae are widely recognized as important factors controlling SOM dynamics, the quantification of contributions of these pathways has not been possible thus far. Our work creates a benchmark in such quantifications, and enables explicit separation

of mycorrhizal impact from climate factors in the topsoil SOC formation process, which can be applied in a broad range of ecosystems.

The dynamics of decomposition and accumulation of labile and recalcitrant litter compounds is shaped by the dominance of arbuscular or ectomycorrhizal plants in vegetation: if plant litter is decomposed in EM-dominated vegetation, the accumulation of recalcitrant components of that litter in soil is twice as high as in soils of ecosystems dominated by AM vegetation. This

difference is likely to strongly affect pathways of accumulation of soil carbon. We conclude that mycorrhizal traits are an important driver of soil carbon dynamic which impacts should be examined mechanistically and quantitatively when estimating future terrestrial carbon storage and predicting impacts of climate change.

**Appendix A: Methodological details of Yasso model structure**

The Yasso model represents the decomposing plant litter as five pools of soil carbon compounds, where each recalcitrant group

has its specific decomposition rate (independent from litter type and the initial amount of the composition) (Liski et al., 2005; Tuomi et al., 2011a). It presents the litter decomposition process as a system of linear differential equations (A1):

$$x'(t) = A(C)x(t) + b(t), \ x(0) = x0 \tag{A1}$$

where, $x(t)$ is a vector describing the mass of individual carbon pools as a function of time $(t)$; $x(0) = x0$ represent the initial amount of each carbon fraction; $b(t)$ is the litter input; $A(C)$ is a matrix describing the total decomposition as a function of

climatic conditions $(C)$, where the diagonal values represent the fraction of C being removed from the pool and the non-diagonal terms specify the amount of C transferred to other pools (Viskari et al., 2020).

The total amount of carbon released from individual WAENH pools is the result of two fluxes: (1) carbon transformation flow from and to other pools, (2) the carbon that is not transferred to other pools but is released to the atmosphere as carbon dioxide. The mass fluxes between the different pools and outside the system are accordingly determined by two parameter sets: $p_{ij}$

represent the mass transportation between pools; $\alpha_i$ represent the total decomposition rate of each pool, i.e. the C mass leaving the pool (the sum of C transfer to other pools and C released into the atmosphere). The total mass flux between two pools is thus a product of these two parameters, e.g. the mass flux from pool A to pool W is $\alpha_A * p_{AW}$.

The total decomposition represented by matrix $A(C)$ within the whole system can be represented as a mathematical equation with mass flow matrix, where parameters $p_{ij} \in [0, 1]$ denote the flows between each pair of WEAN compartments $i$ and $j$, and

$K(C)$ represents the impact of climate on decomposition rate (A2).





$$\mathbf{A}(C) = \begin{bmatrix} -1 & p_{WA} & p_{EA} & p_{NA} & 0 \\ p_{AW} & -1 & p_{EW} & p_{NW} & 0 \\ p_{AE} & p_{WE} & -1 & p_{NE} & 0 \\ p_{AN} & p_{WN} & p_{EN} & -1 & 0 \\ p_H & p_H & p_H & p_H & -1 \end{bmatrix} \cdot \mathbf{K}(\boldsymbol{C}) \tag{A2}$$

In the matrix K(C), each element $k_i(\boldsymbol{C})$ describing decomposition of WAENH is a function of temperature(T), and the annual precipitation(P) modelled in (A3):

$$k_i(\boldsymbol{C}) = \frac{\alpha_i}{J} \sum_{j=1}^{J} \exp\left(\beta_{i1} T_j + \beta_{i2} T_j^2\right) (1 - \exp(\gamma_i P)), \ i \in \{W, A, E, N, H\} \tag{A3}$$

where, βi1 and βi2 are parameters describing the dependency of heterotrophic respiration on temperature, assessed through a Gaussian model (Tuomi et al., 2009); γi is a parameter describing the dependency of heterotrophic respiration of precipitation, assessed through an exponential function (Tuomi et al., 2009). Systematic error in the litter decomposition resulting from litter leaching out of the litter bags was corrected by leaching parameters.

**Appendix B: Methodological details of calibration and databases of litter decomposition data used**

There are three main litter decomposition databases used in original Yasso modelling (Tuomi et al., 2011) and our new model parameterization: CIDET dataset with the measurements from Canada(Trofymow, 1998), LIDET dataset with data from the USA and Central America (Gholz et al., 2000) and Eurodeco (ED) dataset with data gathered from several European research projects (Berg et al., 1991). The distributions of these experimental sites are shown in Fig.B1. Details of these datasets used to parametrize our new model are shown in Table B1.

**Fig.B1 The distribution map of litter bags experiments sites.**

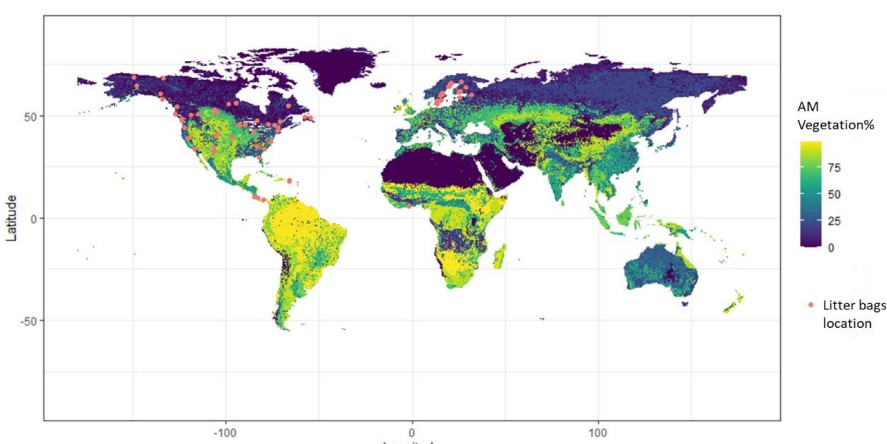

The original Yasso model also uses a dataset with information of SOC accumulation along thousands of years at sites in Finland (Liski et al., 2005) and a large global soil C stock measurements dataset (Zinke et al., 1986) to infer the dynamics of the most stable carbon –Humus pool in soil (see Fig.1). However, concerning our inquiry about impacts of mycorrhizas on dynamics of chemical compounds during plant litter decomposition, and given that the LIDET, CIDET, and ED databases of litter decomposition, used for calibration of our modified YASSO model, store data of 0-10.5 years of decomposition, we assumed no measurable amounts of humus being formed during this time frame. Therefore the mycorrhizal impacts on H related decomposition terms (αH & pNH, see Fig.1) were set to zero.





**Table B1. Dataset description with general environmental conditions**

| Dataset | n | No. of species | Time range (year) | T range (C) | P range (mm) | Elevation range (m) | Mesh size (cm) | Site conditions |
|---------|---|----------------|-------------------|-------------|--------------|---------------------|----------------|-----------------|
| CIDET | 1259 | 12 | 0~6 | -9.8~9.3 | 261~1782 | 48~1530 | 0.25×0.5 | 21 sites, with a broad range of ecoclimate regions cross subarctic, cordilleran, acid and transitional grassland, cool temperate and boreal forest. |
| LIDET | 5900 | 29 | 0~10 | -7.4~26.3 | 150~3914 | 0~2650 | 0.055×0.056 | 27 sites, cover a wide range of climate and biomes: tundra, boreal forest, temperate forest, desert, grassland, humid tropical etc. Including leaf and fine root litter in experiments. |
| ED | 2184 | 5 | 0.55 | 0.2~7 | 469~1067 | 46~350 | 1×1 | Sites locate in boreal and temperate forests. |

**Appendix C: Details of Myco-Yasso parameters and performance**

**Table C1. Posterior and 95% Bayesian credibility intervals (confidence limits) for the Yasso_Myco parameters**

| Parameter | Remark | Unit | Lower limit | Upper limit | Mode |
|-----------|--------|------|-------------|-------------|------|
| $aW$ | Decomposition rate parameter of W | $yr^{-1}$ | 12.111 | 13.906 | 12.834 |
| $aA$ | Decomposition rate parameter of A | $yr^{-1}$ | 1.238 | 1.428 | 1.306 |
| $aE$ | Decomposition rate parameter of E | $yr^{-1}$ | 0.313 | 0.361 | 0.343 |
| $aN$ | Decomposition rate parameter of N | $yr^{-1}$ | 0.137 | 0.197 | 0.134 |
| $pWA$ | Relative mass flows from W to A | - | 0.388 | 0.429 | 0.404 |
| $pWN$ | Relative mass flows from W to N | - | 0.199 | 0.218 | 0.206 |
| $pEW$ | Relative mass flows from E to W | - | 0.891 | 0.989 | 0.961 |
| $b1$ | Temperature dependence of W,A,E | $°C^{-1}$ | 0.059 | 0.066 | 0.063 |
| $b2$ | Temperature dependence of W,A,E | $°C^{-2}$ | -0.002 | -0.001 | -0.001 |
| $bN1$ | Temperature dependence of N | $°C^{-1}$ | -0.004 | 0.006 | 0.004 |
| $bN2$ | Temperature dependence of N | $°C^{-2}$ | -0.003 | -0.002 | -0.003 |
| $g$ | Precipitation dependence of W,A,E | $m\ yr^{-1}$ | -2.234 | -1.859 | -1.956 |
| $gN$ | Precipitation dependence of N | $m·yr^{-2}$ | -2.511 | -1.634 | -2.319 |
| $mAM$ | AM mycorrhiza dependence of W,A,E | $g·m^{-2}·yr^{-1}$ | -0.244 | -0.174 | -0.217 |
| $mEM$ | EM mycorrhiza dependence of W,A,E | $g·m^{-2}·yr^{-1}$ | -0.310 | -0.285 | -0.290 |
| $mN\_AM$ | AM mycorrhiza dependence of N | $g·m^{-2}·yr^{-1}$ | 2.252 | 5.321 | 4.721 |
| $mN\_EM$ | EM mycorrhiza dependence of N | $g·m^{-2}·yr^{-1}$ | 0.333 | 1.461 | 1.233 |






**Fig.C1 Scatter plots of predictions for C loss from plant litter made by original Yasso15 model (grey circles) and predictions made by Myco-Yasso model (blue dots) compared to experimental measurements.**

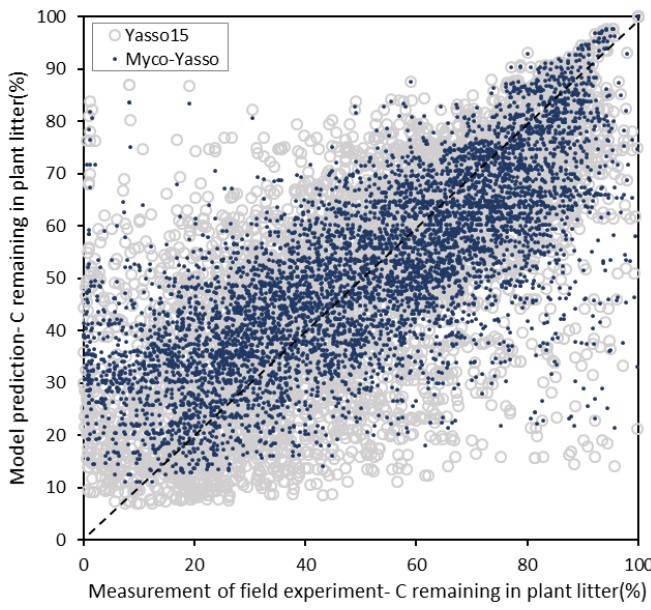


**Fig.C2 Correlation between parameters Myco-Yasso C model. From the most intensive blue colours to most intensive red colours, indicate negative correlations (-1, 0) to positive correlations (0, 1).**

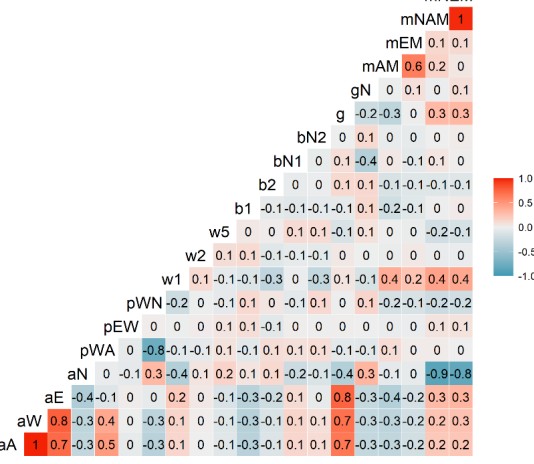




**Fig.C3 Summarized information about model sensitivity of Yasso15 and Myco-Yasso models to individual groups of parameters. The impact of parameters increase by 1% is shown.**

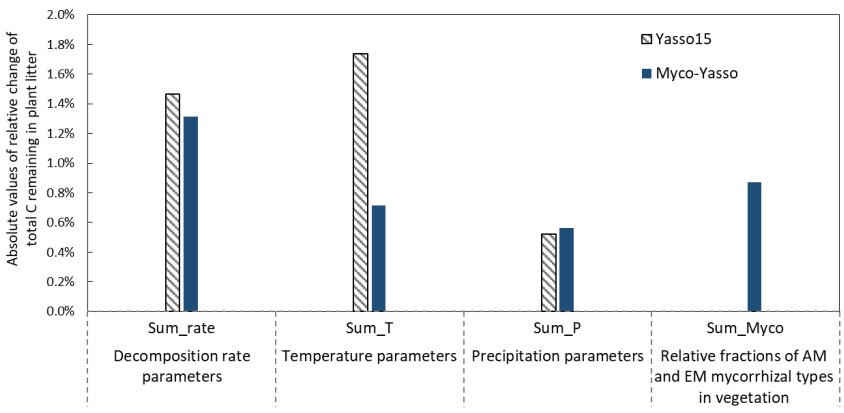

**Fig.C4 Sensitivity of Yasso15 and Myco-Yasso models to1% increase of mass flow parameters. The two most left bars show sensitivity to joint impact of all p-term parameters being increased by 1%. Other bar couples show impact of impact of 1% increase of individual p-terms: pWA – C flux from W to A pool, pEN - C flux from E to N pool, pWN – C flux from W to N pool.**

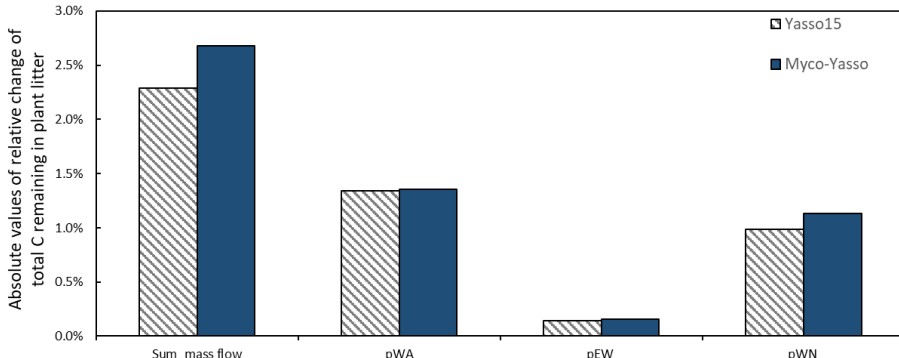

**Fig.C5 Improvement in Myco-Yasso model accuracy compared to the original Yasso over decomposition period using the validation dataset. Bars represent the relative RMSE differences between Yasso15 and Myco-Yasso per period. The line with dots shows the absolute value of the RMSE differences (Yasso15- Myco).**

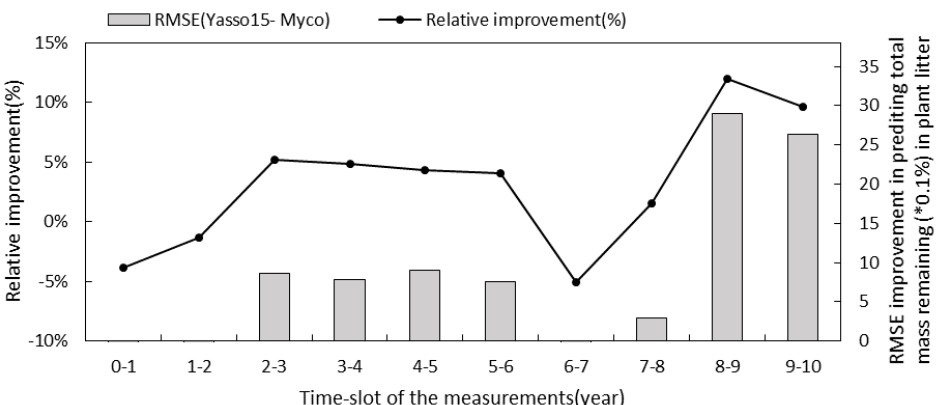



**Appendix D: Experiment on litter decomposition dynamics of different initial litter quality under AM dominant and EM dominant environment**

**Fig.D1 Dynamics of plant root litter decomposition under AM dominant vs EM dominant environment. (a) – loss of total carbon mass from root litter, (b),(c), (d), dynamics of loss of labile carbon components (W – water-soluble C fraction, E – ethanol-soluble C fraction, A – acid hydrolysable C fraction). (e) dynamics of loss of recalcitrant (non-hydrolysable) C fraction (N fraction). The initial WEAN composition of root material is 17%-W, 55%-A, 9%-E, and 20%N (typical for plant roots).**

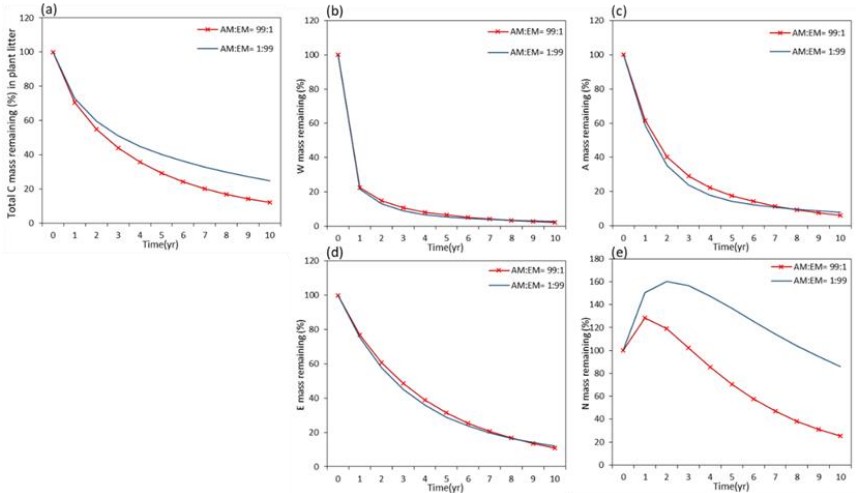

**Fig.D2 Dynamics of plant foliage(leaf) litter decomposition under AM dominant vs EM dominant environment. (a) loss of total carbon mass from root litter, (b),(c), (d) show dynamics of loss of labile carbon component, respectively (b)Water-soluble C fraction, (c)Acid hydrolysable C fraction, (d) Ethanol-soluble C fraction, and (e)dynamics of loss of recalcitrant (non-hydrolysable) C fraction (N fraction). The Initial WAEN composition of root material is 25%-W, 45%-A, 12%-E, and 18%N (typical for plant foliage).**

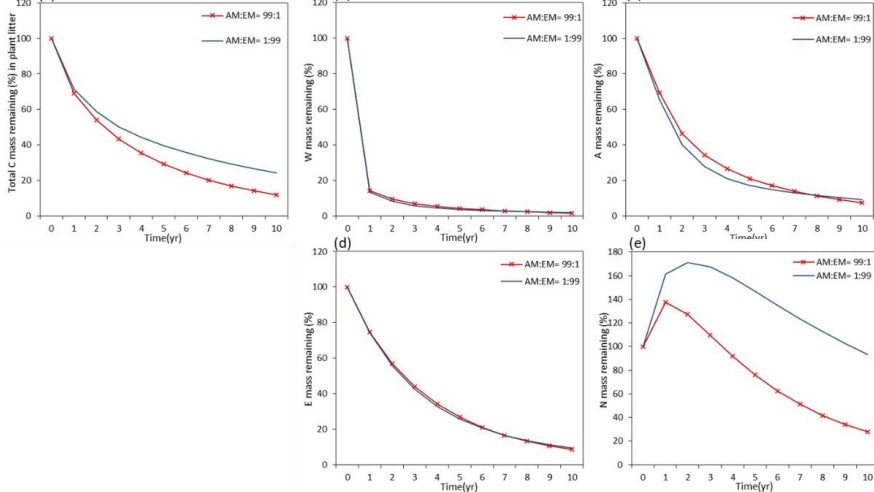

**Code availability**

The initial Yasso15 model is available from the developers repository at https://github.com/YASSOmodel/YASSO15, the code used for calibrating Yasso are available at https://doi.org/10.5194/gmd-2021-273 (Viskari et al., 2021). The extended code for calibrating the models and produce the results, input data and scripts used in this paper is archived on Zenodo at 10.5281/zenodo.5579682 (Huang et al., 2021).



## Data availability

Original litter decomposition data used for this work were provided by the data owners of the different long-term experiments. Please contact them to get access to the data (see Table B1).


## Acknowledgements

This research was supported by the vidi grant 016.161.318 issued to NAS by the Netherlands Organization for Scientific research and China Scholarship Council (CSC, grant No. 201706040071). We gratefully acknowledge the researchers whose data and model are used in the analyses.

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
