# Peer review of "Implementation of mycorrhizal mechanisms into soil carbon model improves the prediction of long-term processes of plant litter decomposition"

_Biogeosciences, 2021_

## Author Comment (AC1)

**Supplements:**

**Implementation of mycorrhizal mechanisms into soil carbon model improves the prediction of long-term processes of plant litter decomposition**

Weilin Huang[1], Peter M. van Bodegom[1], Toni Viskari[2], Jari Liski[2], and Nadejda A. Soudzilovskaia[1,3]

[1]Environmental Biology, Institute of Environmental Sciences, Leiden University, Einsteinweg 2, 2333CC Leiden, the Netherlands;
[2]Finnish Meteorological Institute, Helsinki, 00101, Finland
[3]Centre for Environmental Sciences, Hasselt University, Martelarenlaan 42, 3500 Hasselt, Belgium

[Figure]

*Fig.S1 Dynamics of plant foliage litter decomposition as a result of variation in dominance of AM vegetation (0~1). (a) loss of total carbon mass from root litter, (b),(c), (d) show the dynamics of loss of labile carbon components, being (b)Water-soluble C fraction, (c) Acid hydrolysable C fraction, (d) Ethanol-soluble C fraction, respectively, and (e) dynamics of loss of the recalcitrant (non-hydrolysable) C fraction (N fraction). The initial WAEN composition of decomposition material is 25%-W, 45%-A, 12%-E, and 18%N (typical for plant foliage).*

[Figure]

Fig.S2 Dynamics of plant foliage litter decomposition as a result of variation in dominance of EM vegetation (0~1). (a) loss of total carbon mass from root litter, (b), (c), and (d) show the dynamics of loss of labile carbon components, being (b)Water-soluble C fraction, (c) Acid hydrolysable C fraction, (d) Ethanol-soluble C fraction, respectively, and (e) dynamics of loss of recalcitrant (non-hydrolysable) C fraction (N fraction). The initial WAEN composition of decomposition material is 25%-W, 45%-A, 12%-E, and 18%N (typical for plant foliage).

---

## Author Response (AR2)

**Feedback to reviewers' comments**

Dear reviewers, dear editors,
Thank you very much for your appreciation of our manuscript and the constructive comments and suggestions. This is the updated version of rebuttal compares to responses shared during open discussion period. In the text below, reviewers' comments are numbered in order and marked in *italic*. We have addressed them one by one, and our responses are highlighted in **bold**. The line numbers in this text refer to the lines in the revised manuscript (marked-up version). To ease the reviewing, we have highlighted all the changes in the manuscript file with blue colour.
With best wishes,
Weilin Huang and co-authors

**Response to Reviewer 1**
*R1.0. Huang and co-workers propose an improved version of the C cycling model Yasso that includes the role of mycorrhizal fungi in litter decomposition. It is recognized that mycorrhizal fungi play a major role in decomposition, and that distinguishing EM and AM fungi can increase the level of mechanistic detail in C cycling models, so the topic is timely and suitable for the readership of Biogeosciences. The manuscript is mostly clear and the figure and tables provide a good summary of the findings. I have, however, some conceptual and technical concerns, in addition to comments on the text and presentation.*

**Re R1.0: Thank you for your overall positive feedback as well as for the detailed suggestions to improve the manuscript. Our work incorporated the mycorrhizal fungi impacts as a driver of decomposition conversion rates, explicitly decoupled from climate. The mycorrhizal impact is based on the decomposition of chemical components of different decomposability (WAEN) and their mass flows in the litter. Like in most other process-based ecological models, in our model, mycorrhiza activities are not modelled directly as enzyme activities, but we represent the overall mycorrhizal impact as a function of the mycorrhizal abundance in the vegetation (reflecting the biomass-ratio hypothesis; Grime et al. (1988)) and well-established impacts on different pools of decomposability, as incorporated in many soil carbon models, including original versions of Yasso. The model was calibrated and validated using in-situ measurements of the dynamics of WAEN fractions with decomposing litter material over time. We have addressed the referee comments point to point as indicated below. Our responses are highlighted in bold.**

*Main concerns*

*R1.1. The main result (in my view) is that EM fungi slow down decomposition compared to AM fungi, based on the sign of the m coefficients in Eq. (2). This equation is not mechanistic, meaning that it does not model fungi per se, but rather it accounts for the effect of GPP on decomposition rates, assuming that such an effect is mediated by fungi. Fungal effects come into play because some species are associated with EM and others to AM, so the GPP effect varies from negative to positive. However, it is not possible with this formulation to attribute the altered decomposition rates to fungi. It is possible that decomposition is just faster or slower—for given litter type—depending on vegetation type. For example, needles of pine trees (often associated with EM) have high C:N and decompose relatively slowly, so that they can negatively affect the decomposition of the incubated litter by immobilizing nutrients or capturing labile C and nutrients percolating through the forest floor. In other words, I wonder if the interesting results found here are actually an indication of site effects mediated by plant community composition in general, rather than mycorrhizal fungi in particular. Without a clearer mechanistic link between occurrence of AM or EM fungi and decomposition, it is difficult to attribute these effects to fungal activity.*

**Re R1.1: We agree that mycorrhizal environments are considered here to represent the overall impacts of mycorrhizal activities. As summarized in the reply to R1.0., we accounted for the mycorrhizal fungi impacts on the conversion rate in the model, similar to what other soil C models do. These impacts (for a given abundance of mycorrhizal fungi) are represented by the *m* parameters that are explicitly set being independent from GPP (making decomposition go slower or faster depending on the presence of mycorrhizal fungi). In this way, mycorrhizal presence and identity modify the actual decomposition rate of litter with a particular decomposability.**

**In addition, decomposability itself is indeed likely to be affected by the composition of the vegetation and of plant species identity in particular (as indicated by the reviewer). To account for this dependency, litter decomposability was taken as inputs of the model (information comes from litter bag experiments). Please note that the decomposition results examined here were also from litter bag experiments. The model shows how the local environment would affect the decomposition taking place in the bag, separately accounting for the impacts of mycorrhizal fungi and the climate. We will clarify this line of reasoning throughout the text.**

*R1.2. Calibration/validation. L190: it is not clear how the validation data was selected— 20% of data points within one decomposition time series, or 20% of the time series? If the validation was done on data points within the same decomposition time series on which also calibration was performed, it would not represent a very strict test, as points within*

*a time series are well correlated. In Table 2, it is shown that RMSE actually increase in some of the improved versions of the model, which have more parameters. With a higher degree of freedom, I would expect a reduction in RMSE, unless the model is constrained in such a way that the 'improvement' is actually counterproductive and decreases model performance.*

**Re R1.2: Firstly, the validation dataset was selected as 20% of the time series. We will make it clear in the manuscript, rephrasing the current Lines 194-195, "using 80% of the decomposition time series randomly drawn from the dataset for calibration and the remaining 20% of the decomposition time series for validation". Secondly, for Table2, the RMSEs were shown for the validation dataset containing data not used in calibration. However, the AIC and BIC were based on the performance of the calibrated dataset, which used 80% of the dataset. We will specify this in the Methods section to avoid confusion, rephrasing the current text starting from Line 199: "We use root mean square error (RMSE) from the 20% validation dataset, and Akaike information criterion (AIC) and Bayesian information criterion (BIC) based on the 80% data used for calibration as the criteria for comparing the relative quality of the models".**

**It is true that if we would have used the same data for calibration and validation, we would expect to see the RMSE decrease with the increasing parameter number. However, part of that decrease would be due to over-parameterization which would cause the model to perform worse when tested with data not part of the calibration. To avoid biased comparisons, we based the RMSE on the validation dataset.**

*R1.3. Yasso is based on pools defined according to chemical characteristics, so it should be possible to test the model against lignin data, which are often available together with mass loss data in litter decomposition datasets. I would suggest using also lignin data, as the model is currently poorly constrained using only total mass loss for calibration.*

**Re R1.3: As summarized in the reply to R1.0., the model is constrained with measurements of litter composition from the initial stage till the end of the decomposition. Lignin is indeed available in some of the databases, but it is not available in all datasets. Instead, we used data on the WAEN fractions, available in the datasets (next to datasets that indeed only had a total mass loss). We modified the text to address this issue, see current Line 141: "Chemical composition data consists of the initial composition of litter in terms of WAEN fractions which were measured for each site. This data, together with other environmental data, were used for initializing the model. In addition, for the ED dataset, WAEN components had been determined during the decomposition process and at the end of the decomposition. In addition, all datasets were supplemented with site-specific estimates". These WAEN fractions also**

**indirectly account for lignin as the majority of the constitutions in N pool consists of lignin.**

*R1.4. Section 3.3: if I understand correctly, this figure is drawn by assuming the same baseline model parameters and then adding mycorrhizae in Myco-Yasso. But this can increase or decrease decomposition rates, depending on whether the m coefficients in Eq. (2) are positive or negative and on the proportion of AM vs. EM fungi. So the shift in the mean mass remaining can be guessed by looking at the sign of the AM or EM effect. The reduction in variance could be linked to the change in mean, and might not be an intrinsic property of the Myco-Yasso model. My impression is that a more meaningful comparison could be done by setting parameters after fitting the original and modified models to the same dataset, so the mean mass loss is constrained. Then the change in variance can be attributed to the model modifications, not the overall different decomposition rates.*

**Re R1.4: This sensitivity analysis reflects the general model prediction variability as a result of the uncertainty in parameters. And indeed, the original and modified models were fitted to the same dataset, but the two models were showing different mean values of prediction. The variation in predicted mean values compared to the original and modified model is a result of the two models' distinct structures. Because their abilities to predict litter decomposition under different ecological environments vary, the modified model will have different predictions compared to the original model under different climate conditions. However, given that we base our analysis on the change in decomposition upon a change in mycorrhizal abundance, these differences in mean values do not affect our sensitivity analysis results.**

Other comments

*R1.5. L26: the model is limited to litter decomposition, so I would not conclude that results are relevant for soil C modelling*

**Re R1.5: We agree with this comment, and will re-phrase the statement stressing that we assessed the litter decomposition process in topsoil profiles across 10-years, examining the processes important for initial stages of SOM formation, yet not for long term soil C turnover. Further, we mentioned in the main text that more work needs to be done to include the most recalcitrant compartment of soil defined by the Yasso model as "humus" and address the fate of stable, mineral-associated soil C, the ultimate pool of soil-sequestered C (see discussion section4.4 in current Lines 425-440).**

*R1.6. L62: selective uptake of N does not necessarily increase recalcitrance—it just leaves more C behind. What is the mechanism for increased recalcitrance?*

**Re R1.6: With less N, C is bound to more recalcitrant molecules, see for instance the paper 'Carbon availability triggers the decomposition of plant litter and assimilation of nitrogen by an ectomycorrhizal fungus (Rineau et al., 2013)'. We have added this reference to the main text.**

*R1.7. L91: please define "best representation"—according to what criteria?*

**Re R1.7: Please find the description of the model selection criteria in the methods section "We use root mean square error (RMSE) from the 20% validation dataset, and Akaike information criterion (AIC) and Bayesian information criterion (BIC) based on the 80% data used for calibration as the criteria for comparing the relative quality of the models. The conceptualization with the lowest RMSE, AIC and BIC was selected as the optimal model with best performance", previous Lines 194-199 (current Lines 199-202).**

*R1.8. L94: model errors, but also robustness*

**Re R1.8: Thanks for pointing this out, we adopted your suggestion and have changed the sentence in the current Line 93 to "…in terms of model errors, robustness and temporal dynamics".**

*R1.9. L162: for consistency, "WAEN" not "AWEN"*

**Re R1.9: Thank you, we have corrected it to "WAEN" and make sure the whole paper use the same order of "WAEN" for consistency.**

*R1.10. L171: "a model where the magnitude…"*

**Re R1.10: Thank you, we have made the sentence clear by changing it to "Myco-Yasso.v1 – a model where the magnitude of mycorrhizal impact on carbon loss from each of the W, A, E, and N pools differs among the pools…", see current Lines 171-172.**

*R1.11. Eq. (2) and Table 1: units of the m coefficients should be the inverse of the units of GPP, so units in Table 1 are not correct*

**Re R1.11: Thank you, the unit of GPP per unit area is "g m$^{-2}$ yr$^{-1}$", thus the unit of *m* coefficients should be "g$^{-1}$ m$^2$ yr", it has been corrected.**

*R1.12. L212: wouldn't it be more interesting to let these proportions vary to see the effect of EM vs. AM fungi?*

**Re R1.12: We indeed performed the sensitivity analysis by varying AM and EM proportions, but we did not include it in the appendix as we also found that the magnitude of this effect from mycorrhizal proportions on litter decomposition was dependent on the climate conditions (while the sensitivity analysis was performed at global mean climate conditions). We do not have enough space for more discussions on this aspect, but we plan to address this issue in future papers. However, we included an estimation of litter decomposition by varying AM or EM proportions in the supplementary material (Fig.S1&Fig.S2) which might be interesting for some of the readers.**

*R1.13. L213: please use consistently either annum or year as time unit*

**Re R1.13: We have modified the time unit to "yr$^{-1}$" and made sure the units in this paper are consistent.**

*R1.14. L253: large positive residuals would equally show low predictive power, such as in the low EM% case*

**Re R1.14: We would like to emphasise that the residuals are inclined to be negative in these cases. Indeed, the low EM% cases with large positive residuals also show low predictive power. We have added this fact to the sentence by modifying the sentence to "The model had relatively large negative residuals at low values of the AM fractions (AM<10%) and high values of EM fractions (EM>85%), but relatively large positive residuals at low values of EM fractions (EM<10%), which suggest a lower predictive power for these conditions".**

*R1.15. L296: this is an interesting result!*

**Re R1.15: Thank you for your support.**

*R1.16. L353: I would rather say that litter decomposition is one of the most studied and understood aspects of C cycling… much less is known about C stabilization in the mineral soil for example*

**Re R1.16: We agree that there are other less-known aspects of soil C cycling. Still, there are a lot of uncertainties and unknowns about litter decomposition when we consider microbial interactions and global climate changes. Therefore, we have rephrased the sentence as follows: "There are still many uncertainties and unknowns in the temporal dynamics of litter decomposition, even though it is an essential process with the soil C cycle", current Lines 356-357.**

*R1.17 L399: then the mechanism for increased recalcitrance mentioned in the Introduction is the production of N poor and chemically recalcitrant necromass?*

**Re R1.17: This relates to comment R1.6: with less N, C is bound to more recalcitrant molecules. Please see our reply to R1.6.**

*R1.18 L470-471: "i1", "i2", "i" should be subscripts*

**Re R1.18: Thanks for pointing this out, we will pay attention to these symbols and make sure they are in the correct format, indeed subscripts in this case.**

*R1.19 Table C1: are these parameters resulting from calibration of the whole dataset? In L194 it is explained that models are evaluated for different datasets separately, so I was expecting a parameter table for each dataset*

**Re R1.19: It seems that there are some misunderstandings. The parameters in TableC1 are the result of the calibration of the whole dataset including CIDET, LIDET and ED. In the text starting at Line 197 (current manuscript version), we explained that the model performance of RMSE and $R^2$ were evaluated separately "To account for the fact that the data in the different datasets varied in measurement uncertainty and the number of observations". Thus following the calibration for the dataset as a whole, the resulting parameterization set was used to assess the prediction accuracy of each dataset separately. We have modified our descriptions to avoid confusion, see changes in current Lines 194-201.**

*R1.20.  Fig. C2, legend: "parameters of the Myco-Yasso…"*

**Re R1.20: We agree, this is a confusing sentence, and we have changed the legend to "Correlations between parameters of the Myco-Yasso C model …." as suggested.**

**Response to Reviewer 2**

*R2.0. This study investigates variation in litter decomposition across ecosystems of varying plant-mycorrhizal associations by adding a mycorrhizal association effect to the Yasso15 soil carbon decomposition model. The model was parameterized using a Markov chain Monte Carlo approach from a large set of litter decomposition measurements. The model changes are a simple but reasonable approach to incorporating site-level effects of mycorrhizal associations. Interactions between mycorrhizal associations and decomposition are an important area of study in biogeochemistry and these model developments represent a valuable step toward incorporating these processes into models. However, I think care needs to be taken not to over-interpret the results. The model formulation is a simple linear function of mycorrhizal association on decomposition rates, and does not address the mechanisms of mycorrhizal-decomposition interactions. The parameters and their effects on decomposition are determined by statistical parameter fitting and are validated only using statistical measures of fit to total decomposition over time, so interpretations of the resulting model in terms of changes in litter composition over the course of decomposition are not very strongly supported. The actual improvement in RMSE and related measures is very small, which undermines the stated value of the model changes. I think the interpretation of the results and the support for the value of the model developments would be more robust if model predictions were compared to specific observed trajectories of decomposition over time from sites with different mycorrhizal associations and similar climates (rather than statistical measures over the whole dataset that could hide other covariates or effects). The interpretations would be greatly strengthened if they could be compared with actual measurements of litter composition or lignin content over time. As it is, much of the interpretation of the results relating to different modes of decomposition and changes in the relative amount of labile and recalcitrant litter fractions over time for different mycorrhizal associations is based purely on a model that was not constrained using measurements of litter chemical composition over time.*

**Re R2.0: Thank you for providing helpful comments which will serve to improve the reader's experience of the manuscript. Below we address these points one by one following the detailed requests specified by the referee. Our responses are highlighted in bold.**

Specific comments:

*R2.1. Figure 1: It would be helpful to label the blue CO2 arrows with the percentage that is converted to CO2. This can probably be inferred from the labeled green arrows but it's not immediately straightforward what the respired fraction is because there are several green arrows that need to be added up.*

**Re R2.1: Thank you for your suggestion, we have added the label for large $CO_2$ arrows, for small flows we have made it the dotted lines and have added explanations in the captioin.**

*R2.2. Line 145: What chemical composition data were available? Does this mean that the initial composition of litter (in terms of model pools) was measured for each site and used in model initialization?*

**Re R2.2: Yes, this is exactly the case. "Chemical composition data" contains the initial composition of litter in terms of WAEN fractions which was measured for each site, and this data together with other environmental data were used for initializing the model. We modified the text to address this issue, see current Line 141: "Chemical composition data consists of the initial composition of litter in terms of WAEN fractions which were measured for each site. This data, together with other environmental data, were used for initializing the model. In addition, for the ED dataset, WAEN components had been determined during the decomposition process and at the end of the decomposition. In addition, all datasets were supplemented with site-specific estimates ….", also see reply to Reviewer 1 in R1.3.**

*R2.3. Line 149: If plant community composition was available for each site, then why was the map of mycorrhizal associations needed? Wouldn't local measurements of plant community composition be more accurate?*

**Re R2.3: Thank you for spotting this inconsistency. The text in Line 149 should be "…the ecosystem type of each site was carefully checked for consistency with the map", because the plant community composition information was not available for all sites.**

*R2.4. Table 1: Are the "a" parameters here the same as the "alpha" parameter in Equation A3?*

**Re R2.4: Yes, the "alfa" parameters ($a_W$, $a_A$, $a_E$ and $a_N$) in Table 1 are the same terms of "ai" as shown in Equation A3. We have added the description in current Line 477 to make it clear, "…$\alpha_i$ are decomposition rate parameters". We will make sure to use one**

symbol representing "alfa", using "$\alpha$" throughout the text to keep the consistency and avoid confusion.

*R2.5. Line 230: It's not clear what was different about the inputs in Appendix D. Does this mean that the main simulations used measured chemical fractions for each site while the Appendix D simulations used an average chemical fraction? Is there a table or graph somewhere of the chemical fractions from the sites that were used?*

**Re R2.5: For the comparisons in Appendix D, two sets of initial litter composition data were used for representing two typical types of litter material, i.e. roots (17%-W, 55%-A, 9%-E, and 20%N) and leaves (25%-W, 45%-A, 12%-E, and 18%N). In this comparison, we aimed to demonstrate that the mycorrhizal impacts of AMvsEM dominance are not affected by the initial litter types. And global mean values of all types of litters (including leaves and roots, except woody litters) were used in Fig.9. These chemical fractions of initial litter from sites are available in the datasets. We added text to explain that the chemical composition of the initial litter was available, see changes on Line 141, also see reply to R1.3 and R2.2.**

*R2.6. Line 237-238 and Table 2: The differences in RMSE seem very small, and in many cases RMSE was higher in the mycorrhizal model than in the original model. Overall it seems like weak evidence supporting new model developments. It's also hard to understand why AIC and BIC decreased for models that had higher RMSE and more parameters than the original model (like V4, which had higher RMSE for all three datasets but a lower AIC). How can the updated model be a better fit than the original if it had higher error? Maybe there is a better way to show model improvement than these statistics which don't present a very strong case.*

**Re R2.6:  Indeed, the improvement of the performance parameters of the model is relatively small. The main reason for it is that the original YASSO model is predicting soil carbon dynamics with high accuracy (ca 90%). Thus, there is actually little opportunity to further improve the predictive capability of the model. However, the original YASSO model does not explicitly account for mycorrhizal impacts. Instead, it accounts for the entire suite of environmental conditions that include activities of mycorrhizal fungi and climate. Yet, these two drivers of decomposability (climate and mycorrhizal impacts) are principally different in the nature of the imposed mechanisms. Thus, though the original YASSO model was parameterized to provide accurate predictions of litter decomposition, it provides these predictions while not separating individual drivers (a feature not unique for YASSO, indicating the relevance of our study). The main advancement of our new model, is the explicit separation of these two drivers, allowing us to account for climate per se versus alterations in mycorrhizal types (which could for**

instance be the result of land management actions). The examination of formal performance parameters (RMSE, AIC, BIC) has the primary aim of selecting the best model, among the set of alternative models conceptualizing distinct ways of representing the mycorrhizal impact as being mechanistically independent of climate. We will clarify this logic throughout the manuscript by emphasizing this main aim better in the introduction, showing how this separation affects model parameters and subsequently model sensitivity and discussing the consequences of these changes for predictions of litter decomposition.

Yet following the requests to clarify the logic of model selection, we will provide more details on this, accounting for the following aspects: Firstly, the validation dataset was selected as 20% of the time series. We will make it clear in the manuscript in Lines 194-195, "using 80% of the decomposition time series randomly drawn from the dataset for calibration and the remaining 20% of the decomposition time series for validation". Secondly, for Table2, all RMSEs were assessed with the validation dataset containing data not used for calibration. However, the AIC and BIC were based on the performance of the calibrated dataset, which contained 80% of the dataset. We will specify this in methods to avoid confusion, from Line 199: "We use root mean square error (RMSE) from the 20% validation dataset, and Akaike information criterion (AIC) and Bayesian information criterion (BIC) based on the 80% data used for calibration as the criteria for comparing the relative quality of the models. The conceptualization with the lowest RMSE, AIC and BIC was selected as the optimal model with best performance".

Please also note, that our optimal model (V2) had a lower RMSE for all datasets than the original model. Therefore, although differences are not large, we conclude –in combination with the substantially improved BIC and AIC- that the performance of the mycorrhizal model is (marginally) better. Importantly, in doing so, the model represented more closely the complexity of drivers expressed in the soil environment (as explained above).

*R2.7. It would also be helpful to include Pearson's r in Table 2.*

Re R2.7: Thank you for the suggestion. We presented the Pearson's r in text in Lines 235-236 (current Lines 237-238). However, we did not include it in Table 2 given that Pearson's r does not account for differences in degrees of freedom (in contrast to AIC and BIC). Therefore, we considered it an inferior metric to base model selection on.

*R2.8. Figure 8: It's not clear which axis scale (right or left) is used for the bars and which is used for the line*

**Re R2.8: As explained in the caption of the figure, "Bars represent the relative RMSE differences between Yasso15 and Myco-Yasso per period. The line with dots shows the absolute value of the RMSE differences (Yasso15- Myco)". We will add extra legends on the axes to directly link individual axes to the contents.**

*R2.9. Figure 9: Showing some measurements from sites of contrasting mycorrhizal associations to compare with the model here would make a much clearer case for whether this model behavior is realistic and whether it represents an improvement compared to the original model. It would also be useful to show the prediction of the original Yasso15 model on these plots to show how the mycorrhizal model compares to the original.*

**Re R2.9: We agree that it is important to compare the model performance at sites with contrasting conditions. That is also the reason why we did cross-validation using 20% data, and this data was not used for calibration (i.e. the other 80% were used for calibration), see current Lines 194-195. This validation dataset contains data from contrasting mycorrhizal environments. The prediction of the original Yasso15 compared to the new mycorrhizal model is shown in Fig.C1.**

*R2.10. Line 339: It seems like "environment of plant litter decomposition" could refer to any number of processes, from microbial community to litter quality to physical and hydrological effects of the litter layer… It is useful to assess the combined effect of factors that are integrated by different mycorrhizal associations, but it doesn't provide much insight about the underlying processes. I think this makes it somewhat inaccurate to call this a "mechanistic" approach.*

**Re R2.10: Here we are specifically referring to the "mycorrhizal environment of plant litter decomposition". Indeed, mycorrhiza can affect C cycles via three mechanistically distinct pathways of "(1) provisioning dead mycelium as substrate for decomposition, (2) mediating plant litter quality and amounts, and (3) controlling the environment of plant litter decomposition" (see current Lines 340-343, previous Lines 337-339). Our work in this paper focused on one pathway "controlling the environment of plant litter decomposition", which refers to a composite decomposition environment with different types of mycorrhizal vegetation and its impact to the litter decomposition process. We explicitly did not specify the other two pathways because our dataset does not allow examining these pathways. These pathways could not be examined because the decomposition results examined here are from litter bag experiments and thus not products of the local plant community composition. Thus, mycorrhizal impacts on the**

other two pathways were excluded as litter amount and initial litter types were controlled.

As we explained in the rebuttal to R2.6, the main advancement of our model is an explicit separation of the two drivers of decomposition, climate and mycorrhizas; assessing the integral impacts of mycorrhiza through their effects on decomposition. Therefore, we will remove statements on the "mechanistic accounting for mycorrhizal impacts'', and instead clarify the line of reasoning on the model advances, as we explained in the R2.6.

*R2.11. Line 356: It is difficult to measure changes in composition and breakdown of litter components over time, but that does not mean modeling these processes is easy either! In fact, it's often more difficult to model processes that are poorly understood from the observational side because it means there is a weaker theoretical basis for developing a model.*

**Re R2.11: We agree with this point. We will acknowledge that there are difficulties in the modelling process in the discussion. However, it is also extremely challenging to measure the flow between decomposition pools, which we actually can assess by modelling. We will address this point in the discussion section.**

*R2.12. Line 357: I would say litter decomposition, not soil C*

**Re R2.12: We agree that we are looking at the litter decomposition process, and we admit that we did not look into the mineralization process. However, decomposition is an important process within soil C cycling, and particularly during the initial stages of SOM formation. Reviewer 3 also acknowledged this (see under R3.0). We will soften the tone when we mention soil C to make explicit that our study refers to the initial stage of soil C formation.**

*R2.13. Line 379-381: If mycorrhizal associations are tightly correlated with temperature, wouldn't this also affect the calibrated model? How can we know that the model's results in terms of temperature and mycorrhizae effects are not also driven by large-scale covariation between climate and mycorrhizal association? One way to investigate this would be to show observed patterns of decomposition from sites with similar climates and contrasting mycorrhizal associations compared with the model as in Figure 9.*

**Re R2.13: We acknowledge that mycorrhizal associations are correlated to climate, like almost all natural processes are correlated to climate. There is no doubt that climate is a factor driving global vegetation distribution, and it could also potentially affect**

mycorrhizal associations. The original Yasso15 structure only considers climate as a decomposition factor but without potential mycorrhizal impacts, and according to the sensitivity analysis in Fig.6 and Fig.7, we think that potential mycorrhizal impacts were partly accounted for by climate variables in the original Yasso15. In the new model, we account for mycorrhizal impacts separately from climate, while the magnitude of these impacts is scaled to the abundance of mycorrhizal vegetation. Though the latter is undoubtedly driven by climate, it is accounted for as a separate input parameter of the model, thus model formulation-wise being separated from climate. We agree about the need to run the model for similar climate conditions, and this is exactly how the sensitivity analysis was performed: at global mean climate conditions. To explore these patterns further, we are working on a global estimation of the mycorrhizal impacts, employing the same type of analysis. However, it is not within the scope of this model description paper. We look forward to sharing more interesting findings with you in the future.

*R2.14. Line 407-408: The model does not provide mechanistic insights since it just relates decomposition rates to overall site mycorrhizal associations, not to specific underlying processes. And the model addresses litter decomposition, not soil C.*

**Re R2.14: These two issues seem to be already mentioned and have been addressed within the replies to R2.10 and R2.12. Please see our responses to these points.**

*R2.15. Line 409-411: The accumulation of recalcitrant compounds and impacts on labile compounds were model results and were not validated with any measurements of compounds such as lignin or soluble C over time, so I would be wary about interpreting this too confidently.*

**Re R2.15: There seems to be some misunderstanding here. The database used for calibrating the model has information on different recalcitrant compounds over time in terms of WAEN, not only for the initial stage, but over the whole decomposition process till the end of the decomposition. Hence, our model is capable of predicting the differences in recalcitrant compounds at the end of the decomposition stage. Though it has been specified in Appendix B, we feel it is necessary to add more details to describe the database in Line 138 to avoid confusion: 'We calibrated our new model using litter decomposition databases (Appendix B) used in Yasso modelling that included total mass loss and the dynamics of different chemical components over time (Tuomi et al., 2009, 2011b, 2011a)'. Also, see reply to R1.3, R2.2 and R2.17.**

*R2.16. Line 413: The differences in RMSE from Table 1 and Figure 8 were quite small, so it seems like a stretch to say that it "greatly improves the accuracy." The difference was from RMSE of 19.9 to 19.3, or 10.55 to 10.5, which seems barely significant. Or, according to Figure 8, just a couple of percent of RMSE. If there are alternative metrics that show a clearer improvement, it would be helpful to highlight those. And the model predicted to litter decomposition, not SOM dynamics.*

**Re R2.16: We agree with these points. Very similar concerns were raised in R2.6 (RMSE) and R2.12 (litter vs SOM). We will modify the text in Line 413 (current Line 419), according to the plan proposed under R2.6 and R2.12.**

*R2.17. Line 443-444: The differences in recalcitrant compounds are purely a model result, not constrained by any measurements of litter composition over time so I would be wary of drawing this conclusion too strongly.*

**Re R2.17: Please also see the reply to R2.15, R2.2 and R1.3; our model is capable of predicting the differences in recalcitrant compounds at any decomposition stage, as it is calibrated with data of WAEN during the whole decomposition process and not only at the initial stage. Thus, we draw the conclusion using the words in Lines 443-444 (current Lines 449-451).**

*R2.18. Figure C1: It's hard to tell much difference between the two model versions from this figure. Would color coding the dots by mycorrhizal association of each site help to highlight any improvements from adding mycorrhizal effects to the model?*

**Re R2.18: The scatter plots and the 1:1 line should be helpful to compare the performance of the two models. Dots that are closer to the 1:1 line indicate that those model predictions are closer to measurement values. From Fig.C1, we can see that blue dots are more constrained to the 1:1 line compared to the grey circles. This already supports that Yasso-Myco has a better performance than the original model. We are afraid that colour-coding of the dots by mycorrhizal association might not help to reflect more information but might add more distractions. The performance of different associations of the site can be found in Fig.4.**

*R2.19. Fig. D2: I think this caption should say leaf material, not root material*

**Re R2.19: Thank you for pointing out this error. We will correct the sentence in the caption "The initial WAEN composition of leaf material is 25%-W, 45%-A, 12%-E, and 18%-N (typical for plant foliage)".**

**Response to Reviewer 3**

*R3.0. This study added mycorrhizal impacts on plant litter decomposition to the Yasso15 model, calibrated and validated the new Myco-Yasso model using 3 large-scale litter decomposition datasets, and explored the sensitivity, behavior, and broader implications of the new model.*

*In general, this paper was a thorough, thought-provoking, and enjoyable read. The paper was well-written and well-organized, making it easy to understand the approach the authors took to model development and testing. The sensitivity and model behavioral analyses were thorough and left me with very few of the "but what about..." questions that modeling papers usually give me. The role of AM and EM fungi in soil C cycling is an important factor that models have yet to address, and this paper is thus a timely and novel addition that will likely interest the readers of Biogeosciences. Although the authors' model addressed litter decomposition and not soil C cycling, litter decomposition is an important first step in both experiments and models of soil C cycling that has been rigorously documented by litter decomposition studies, and I think the authors did a successful job of placing their model in the broader context of soil C cycling without overstating the capabilities and implications of their model.*

*I agree with the comments made by other reviewers that mycorrhizal effects were represented within the model via the proxy variable of plant cover type, and that this limits the 'mechanistic' interpretation of the model somewhat. The new model did not incorporate microbial biomass or enzyme pools, and therefore cannot theoretically be validated using any measurements of actual microbial variables. As other reviewers have stated, plant cover type likely correlates strongly with both climate and litter chemical characteristics, which confounds the interpretation of the new model as purely representing mycorrhizal effects. I think the authors' approach to modeling mycorrhizal effects is still valid and interesting, but I think this limitation bears more discussion, especially when numerous other soil C models with explicit microbial impacts have been published recently (CORPSE, MIMICS, ORCHIMIC, the Millenial model, to name a few) and have demonstrated that microbes can be represented in models without relying on proxy variables.*

*I have a few additional language edits to suggest, and I imagine the journal's copy-editing service will catch a few more:*

**Re R3.0: Thank you for your careful reading and encouraging comments on our manuscript. Indeed, there are various concerns about the definition of so-called 'mechanistic' in this paper, and we will rephrase these parts to avoid more debates**

**(Please see the rebuttal to R2.6 and R2.10 for details). And thanks for the recommended papers, we will include them in the discussion section.**

*R.3.1.*

*33 – are not is;*

*42 – are not is;*

*131 – space before the dash*

**Re R3.1: Thank you for your careful reading, we have modified the sentences: Line 33 (current Line 35), '…are the most widespread symbiosis on Earth'; We changed the whole sentence on Line 42 (Current Line 44) 'Plant litter decomposition is an important component of soil C cycling and is affected by its chemical composition (Berg and McClaugherty, 2008; Cornelissen et al., 2007), which is …'; Line 131 (current Line 128), 'stage - contribute to the accumulation of mineral associated organic matter MAOM'.**

*R3.2. 142 – "parameterize" is a modeling term that refers to the representation of a complex process as a simplified mathematical relationship between parameters and is not synonymous with "parameter selection" or "parameter tuning." It would be more appropriate here to say "We selected parameters for our new model…"*

**Re R3.2: Thank you for pointing this out, the word should be 'calibrate'. We modified the sentence to 'We calibrated our new model using litter decomposition databases…', current Line 138.**

*R3.3. 355 – The last clause of this sentence is very awkward*

**Re R3.3: This refers to the sentence "The mycorrhizal impacts are likely less visible in the short-term (< 3 years), and detectable effects of the mycorrhizal environment of litter should be assessed over a longer period", in current Line 363. There are two typos in the last clause. It should be "and detectable effects of the mycorrhizal environment ON litter DECOMPOSITION should be assessed over a longer period."**

*R3.4. 407 – quantitative not quantitatively*

*413 – The paper is careful elsewhere not to overstate the improvement in model performance generated by the new changes; I think the word "greatly" is not appropriate here.*

**Re R3.4: We modified the sentences in Line 407 (current Line 413), 'Our modelling exercises provide a quantitative examination…' and deleted the word 'greatly'.**

*R3.6. 421 – This paragraph doesn't flow as well as the rest of the paper and is somewhat difficult to get through; it could use another pass-through for sentence clarity and concision.*

**Re R3.6: Thank you for sharing your reading experience, we will polish the paragraph and the rest of the paper to give a better reading experience.**

**Response to Reviewer 4**

*R4.0. In the paper "Implementation of mycorrhizal mechanisms into soil carbon model improves the prediction of long-term processes of plant litter decomposition", authors Huang, van Bodegom, Viskari, Liski, and Soudzilovskaia added effects of arbuscular (AM) vs ectomycorrhizal (EM) dominance to the soil carbon model, Yasso15. They selected a model where mycorrhizal dominance differentially affected labile and recalcitrant litter pools, compared predictions of this model to long-term litter decomposition data, and evaluated how AM- versus EM-dominated systems would be expected to differ in litter decomposition dynamics. The mycorrhizal model provided a better fit to the data and it was found that EM systems exhibit slower decay of recalcitrant litter fractions. To me, the most interesting finding was that the mycorrhizal model showed much lower sensitivity to climatic factors (i.e. temperature), suggesting that climatic effects are confounded by vegetation dynamics. Overall, this was a nice paper and a useful development of a model that can be applied at the global scale. But I do have some concerns about the framing and model evaluation.*

*A great deal of the motivation for this study seems to be a better mechanistic representation of mycorrhizal effects. The authors criticize that current models "treat mycorrhizal impacts…as a black box" and describe different potential pathways of mycorrhizal effects. Yet, the approach taken in this study is to add rate-modifying parameters to a linear first-order decomposition model. I find this to be a very useful and reasonable approach to leveraging mycorrhizal information to improve predictions, but I don't see how this analysis advances our mechanistic understanding of mycorrhizal effects in a meaningful way. The authors should consider re-framing with this point in mind. Especially as I see other reviewers have touched on this same point.*

*It is unclear how much of an improvement the mycorrhizal model represents. By RMSE, it appears that the original Yasso15 model is better than three of the four tested Myco-Yasso models (depending on the evaluation dataset) despite having 2-8 fewer parameters. And the change in the mean mass remaining for the case study (fig. 5) is very minor (~5% less*

*mass after 10 years). That would be a difference of k = 0.15 y-1 vs 0.18 y-1 in a single exponential decay model. I definitely think there is value in these mycorrhizal models (for example if climate and mycorrhizal dominance are more decoupled in the future), but I think the authors could make a better case for the value of the proposed changes.*

**Re R4.0: Thank you for your overall positive feedback as well as for the detailed suggestions to improve the manuscript. The points of the mechanistic model and a small improvement in the model predictive power coincide with the R2.10 and R2.6 comments of Reviewer 2. Please see our reply to these points.**

**We have addressed other inquiries in the replies to comments point to point as indicated below. Our responses are highlighted in bold.**

Other comments

*R4.1. Line 24: How does an overestimation of climate impacts result from exclusion of mycorrhiza-induced mechanisms? This is elaborated on in the main text, but is missing context here in the abstract.*

**Re R4.1: We will provide more details in the abstract, changing the sentence in Line 24 (current Line 25) to 'A sensitivity analysis of litter decomposition to climate and mycorrhizal factors indicated that ignoring the mycorrhizal impact on decomposition will lead to an overestimation of climate impacts on decomposition dynamics.'**

*R4.2. 40: Throughout the paper, it is not totally clear what is meant by "controlling the decomposition environment" and how the current study focuses on this. Doesn't the model integrate all potential mycorrhizal effects?*

**Re R4.2: Mycorrhiza can affect C cycles via three mechanistically distinct pathways of "(1) provisioning dead mycelium as substrate for decomposition, (2) mediating plant litter quality and amounts, and (3) controlling the environment of plant litter decomposition" (see Lines 337-339, current Lines 341-342). Our work in this paper focused on one pathway only "controlling the environment of plant litter decomposition", which refers to a composite decomposition environment with different types of mycorrhizal vegetation and its impact to the litter decomposition process. We could not specify the other two pathways as our calibration and validation datasets were based on litter bags experiments in which litter amount and initial litter types were controlled. We will further clarify this in the main text, while combining the comments in R2.15 and R4.8: current Line 138 changed to "We calibrated our new model using litter decomposition databases (Appendix B) used in Yasso modelling that**

included total mass loss and the dynamics of different chemical components over time (Tuomi et al., 2009, 2011b, 2011a)". Also see reply to R2.10 and R4.9.

*R4.3. Fig. 1: What are the time units for the fluxes % yr-1?*

**Re R4.3: The fluxes of '%' are fixed rates/percentages of carbon transformation per time being modelled, as it comes with the decomposition rate, which means that the fluxes are with a unit of yr$^{-1}$. We didn't add it to the figure considering the limited space in the figure, but we have specified the unit in the caption of Fig.1 to make this clear.**

*R4.4. 170: What is the justification for these four particular models and what hypotheses do they represent? This could be better motivated.*

**Re R4.4: As described in Lines 171-177, each model scenario represents a possible conceptualization of mycorrhizal impact, explicitly separated from climate impacts. The Yasso model assumes that different chemical components in the litter decompose with distinct rates of mass flows to other pools and to the atmosphere. We tried different scenarios: Myco-Yasso.v1 is based on the assumption that mycorrhiza impact all these different recalcitrant pools differently, and in Myco-Yasso.v3 we assume the impact is the same for all pools. For Myco-Yasso.v2, we assume that mycorrhiza has a similar magnitude when affecting WEA pools while affecting the N pool differently. This assumption relates to previous findings of Yasso that climate factors have similar impacts on WEA pools, but are different for the N pool. For Myco-Yasso.V4, we assume that mycorrhiza could only affect the most recalcitrant pool of N. We will add these details in the Methods section to describe the assumptions of different model versions (see current Lines 171-181).**

*R4.5. 275: Here and throughout, I recommend just spelling out these different pools (N, W, A, E) or giving a more intuitive abbreviation.*

**Re R4.5: These abbreviations have been defined in Lines 105-108 (current Lines 103-105), and are coherent with all other Yasso papers. The only annoyance would be the N pool, as 'N' is a widely acknowledged abbreviation for nitrogen, while it refers to the Non-hydrolysable pool here. We will remind the reader about this at strategic places in the paper.**

*R4.6. 295: This is a very interesting finding. I do wonder if other confounding factors may still be folded in with mycorrhizal dominance or climate variables (e.g. biome boundaries, soil orders, etc.).*

**Re R4.6: Thank you for your support. We were also interested in this finding, and we are working on a global estimation of the mycorrhizal impact which could probably reveal more insights. However, it is not within the scope of this model description paper, and we look forward to sharing more interesting findings with you in the future.**

*R4.7. Fig. 8. It may be useful to show R2 which is a slightly more intuitive measure of model fit. Also, is there a measure you can use that corrects for the number of model parameters?*

**Re R4.7: We agree that $R^2$ is a way to compare model fit, but it is not suitable in our case as our models had a different number of parameters which is not corrected for in $R^2$. The same applies to Pearson's r, also see reply to R2.7. Instead, both AIC and BIC include a penalty for an increasing number of estimated parameters. You can see both AIC and BIC showing the lowest value in the case of Model.V2 which supported our model selection.**

*R4.8. 323: In addition to total mass loss, confronting models with actual litter fraction data would be extremely valuable for validation. Are these data available for any of the individual experiments?*

**Re R4.8: Thank you for checking this issue which has also been raised by Reviewer 2. As we discussed in the rebuttal to R2.2, R2.15 and R2.17, models were calibrated using the actual measurements of litter decomposition. These data included both the total mass loss information as well as WAEN fractions dynamics over time. Though it has been specified in Appendix B, we feel it is necessary to add more details to describe the database in current Line 138 to avoid confusion: "We calibrated our new model using litter decomposition databases (Appendix B) used in Yasso modelling that included total mass loss and the dynamics of different chemical components over time (Tuomi et al., 2009, 2011b, 2011a)".**

*R4.9. 339: "Earlier works did not explicitly differentiate between these pathways". Maybe I am missing something crucial, but I am not sure how the current study differentiates between these pathways.*

**Re R4.9: In the sentence which you quoted, we aimed to emphasize that mycorrhiza can affect C cycles via three mechanistically distinct pathways of "(1) provisioning dead mycelium as substrate for decomposition, (2) mediating plant litter quality and**

**amounts, and (3) controlling the environment of plant litter decomposition" (see Lines 337-339, current Lines 341-342). Our work in this paper focused on one pathway, i.e. "controlling the environment of plant litter decomposition", which refers to a composite decomposition environment with different types of mycorrhizal vegetation and its impact to the litter decomposition process. We did not specify the other two pathways because we used litter bag experiments in which litter amount and initial litter types had been controlled. Also, see reply to comments R2.10 and R4.2.**

*R4.10. 408: This study looked at litter decomposition specifically. Extending these results to discuss conservation of soil C generally seems like an over-reach. Especially given ideas that litter decay and soil C formation may be positively correlated (Cotrufo et al. 2013; GCB).*

**Re R4.10: Although our model addressed litter decomposition and not soil C cycling, "litter decomposition is an important first step in both experiments and models of soil C cycling that has been rigorously documented by litter decomposition studies"(cited from Reviewer 3). And we thank you for suggesting the interesting paper, but in this sentence, the results refer to the fact that the soil accumulation of most recalcitrant component N in the litter is different in AM vs EM-dominated environments. We assume that this most recalcitrant component is an important source/origin of recalcitrant carbon compounds in the soil. However, we will make sure to be specific whenever we mention the soil C cycle.**

*R4.11. 427: I am unclear how the accumulation of non-hydrolyzable material somehow supports a hypothesis about mineral stabilization of soil C (as this was not addressed in the current study). Also, yes, there is some discussion about different soil C pathways in the literature, but not much evidence that lignin is a main component of mineral-associated soil C. Altogether, the speculation in this paragraph may be beyond the scope of the current paper.*

**Re R4.11: This comment refers to the sentence "While our work does not address the pathway of formation of minerally stabilized carbon, it provides insights into the important processes preceding C mineral stabilization, as we examine the long term processes of formation of labile C pools potentially available for microbial uptake and the development of recalcitrant plant litter pools potentially forming MAOM by binding to mineral particles". We did not aim to support or falsify the hypothesis about mineral stabilization of soil C.  Instead we aim to link to this hypothesis, highlighting that we examined the long term processes of the formation of labile C pools and the development of recalcitrant plant litter pools, and we consider the products of this long-term litter decomposition process as the potential source for mineral associated C**

formation. We considered it interesting to show the potential links of our work to the emerging concepts of mineral stabilization of soil C, but we agree that it is necessary to highlight the hypothetical nature of this link. Thus, we will rephrase the sentence to "While our work does not address the pathway of formation of minerally stabilized carbon, it provides insights into the important processes preceding C mineral stabilization, as we examine the long term processes in labile C pools that are potentially available for microbial uptake and the development of recalcitrant plant litter pools that potentially form MAOM by binding to mineral particles".

*R4.12. Lastly, the work of Sulman and co-authors is highly relevant to the current study, but I did not see any mention of these works:*

*Sulman, B. N., Brzostek, E. R., Medici, C., Shevliakova, E., Menge, D. N. L., & Phillips, R. P. (2017). Feedbacks between plant N demand and rhizosphere priming depend on type of mycorrhizal association. Ecology Letters, 20(8), 1043–1053. https://doi.org/10.1111/ele.12802*

*Sulman, B. N., Shevliakova, E., Brzostek, E. R., Kivlin, S. N., Malyshev, S., Menge, D. N. L., & Zhang, X. (2019). Diverse Mycorrhizal Associations Enhance Terrestrial C Storage in a Global Model. Global Biogeochemical Cycles, 33(4), 501–523. https://doi.org/10.1029/2018GB005973*

**Re R4.12: Thank you for suggesting these interesting papers. One major reason that we did not mention these two papers is that we are not looking at the nitrogen cycle. When specifying nitrogen in the system, it will need more information to represent microbial activities to constrain the model. However, this kind of detailed information is not always available for global modelling. Our model considers the mycorrhizal impact as an integrated function of the environment on the litter decomposition process, which includes all possible chemical and microbial impacts as induced by mycorrhiza without specifying these characteristics. But, indeed, it is important to mention more related works, as also indicated by R3.0 and we have included relevant other models in the discussion (see current Lines377-380).**

**References**

Grime, J. P., Mackey, J. M., Hillier, S. H. and Read, D. J.: Mycorrhizal infection and plant species diversity, Nature, 334(6179), 202–202, doi:10.1038/334202b0, 1988.

Rineau, F., Shah, F., Smits, M. M., Persson, P., Johansson, T., Carleer, R., Troein, C. and Tunlid, A.: Carbon availability triggers the decomposition of plant litter and assimilation of nitrogen by an ectomycorrhizal fungus, ISME J., 7(10), 2010–2022,

doi:10.1038/ismej.2013.91, 2013.